# Single-dispersed polyoxometalate clusters embedded on multilayer graphene as a bifunctional electrocatalyst for efficient Li-S batteries

Jie Lei[1], Xiao-Xiang Fan[1], Ting Liu[1], Pan Xu[1], Qing Hou[1], Ke Li[1], Ru-Ming Yuan[1], Ming-Sen Zheng ID [1], Quan-Feng Dong ID [1✉] & Jia-Jia Chen ID [1✉]

The redox reactions occurring in the Li-S battery positive electrode conceal various and critical electrocatalytic processes, which strongly influence the performances of this electrochemical energy storage system. Here, we report the development of a single-dispersed molecular cluster catalyst composite comprising of a polyoxometalate framework ($[Co_4(PW_9O_{34})_2]^{10-}$) and multilayer reduced graphene oxide. Due to the interfacial charge transfer and exposure of unsaturated cobalt sites, the composite demonstrates efficient polysulfides adsorption and reduced activation energy for polysulfides conversion, thus serving as a bifunctional electrocatalyst. When tested in full Li-S coin cell configuration, the composite allows for a long-term Li-S battery cycling with a capacity fading of 0.015% per cycle after 1000 cycles at 2 C (i.e., 3.36 A g$^{-1}$). An areal capacity of 4.55 mAh cm$^{-2}$ is also achieved with a sulfur loading of 5.6 mg cm$^{-2}$ and E/S ratio of 4.5 μL mg$^{-1}$. Moreover, Li-S single-electrode pouch cells tested with the bifunctional electrocatalyst demonstrate a specific capacity of about 800 mAh g$^{-1}$ at a sulfur loading of 3.6 mg cm$^{-2}$ for 100 cycles at 0.2 C (i.e., 336 mA g$^{-1}$) with E/S ratio of 5 μL mg$^{-1}$.

[1] State Key Laboratory for Physical Chemistry of Solid Surfaces, Department of Chemistry, College of Chemistry and Chemical Engineering, iChem (Collaborative Innovation Center of Chemistry for Energy Materials), Innovation Laboratory for Sciences and Technologies of Energy Materials of Fujian Province (IKKEM), Xiamen University, Xiamen, Fujian 361005, China. ✉email: qfdong@xmu.edu.cn; JiaJia.Chen@xmu.edu.cn

Lithium–sulfur (Li–S) batteries have been considered as the preferred choice for next-generation high-energy-density storage systems, attributing to the low cost and eco-friendliness of sulfur, as well as its extremely high theoretical specific capacity of 1675 mAh g$^{-1}$ [1]. The sulfur redox reactions are obviously crucial in Li–S batteries that commonly involve the complicated multiphase transformation processes between elemental sulfur and lithium sulfides ($Li_2S$) accompanied by a series of dissoluble lithium polysulfide intermediates (LiPSs). The LiPS dissolution and deposition are believed to be helpful to facilitate the complete utilization of the insulating sulfur/$Li_2S$ particles via chemical comproportionation and subsequently redistribute the sulfur/$Li_2S$ with easier access to electrons and ions[2,3]. However, the diffusion of LiPSs in the electrolyte also leads to several adverse impacts, especially the shuttle effect, which would induce the severe self-discharge, anodic corrosion of lithium metal, and low coulombic efficiency[4].

During the last decade, enormous efforts have been made to design and optimize functional sulfur composites, typically by employing various host materials, such as porous carbon materials[5–7], metal oxides[8], metal sulfides[9], and metal nitrides[10,11], to physically or chemically trap the LiPSs and suppress the shuttle effect. Functional binders[12,13] and freestanding electrodes[14] are also able to retard the dissolution loss of LiPSs. Moreover, modified separators[15,16] and solid electrolytes[17] are also developed to alleviate the dissolution and diffusion of LiPSs. Beyond these, electrocatalytic strategy has been proposed in recent years to effectively mitigate the dissolution and accumulation of LiPSs by boosting the LiPS conversion[18–20]. The catalytic activity for sulfur conversion is directly relevant to the exposed active sites that involve the aforementioned strong binding and trapping of LiPS species, rapid charge (electron and ions) transfer within the host framework. Indeed, varieties of adsorption materials reported previously have been proved to synchronously exhibit the electrocatalytic effect on the LiPS redox reactions[21,22]. Besides, single-atom catalysts (SACs)[23–29] have attracted the researchers' attention, which could exhibit excellent catalytic activity due to their atomically dispersed active centers. For instance, monodisperse cobalt atoms[30,31] could greatly enhance the absorption and kinetic reaction process of LiPSs, thus improving high-rate performance and cycling stability of Li–S batteries. However, some crucial challenges remain to be addressed[32–36]. Due to the high-surface free energy, SACs are highly unstable and tend to aggregate, especially with high loading of active metal. Effective supports with specific binding sites should be developed to stabilize the single-metal atoms. In addition, the understanding of the intricate SAC-support interactions and binding modes is important to bridge the structure–activity gap between experimental and theoretical simulation.

Polyoxometalates (POMs) are a variety of atomically well-defined metal-oxide clusters that can serve as the ideal molecular systems to explore the anchoring of single-metal atoms[37]. Particularly, the unsaturated binding sites in the lacunary POMs are ideally suitable for permanent active metal-atom coordination and limited leaching and loss of the active metal centers[38]. The coordination environment and electronic structure of active metal atoms could be determined clearly, which is crucial to gain atomic insights into the fundamental understanding of catalytic mechanism and the corresponding structure–activity relationships. Moreover, POMs offer multifunctionality owing to its fascinating properties, including the multi-electron redox capability and Lewis acidity/basicity, as well as the structural diversity and compositional adjustability[39,40]. These have been exploited for promising applications in the field of electrocatalysis with unexpected catalytic activity[41]. For example, POMs exhibit attractive catalytic behaviors to oxygen and sulfur species. The Ag(I)-substituted polyoxometalate cluster, $K_3[H_3Ag^IPW_{11}O_{39}]$, has been developed to effectively improve the stabilization and conversion kinetics of LiPSs, in which the Ag(I) atom can serve as a Lewis acid site to coordinate with the S moieties in LiPSs, while the terminal oxygen atoms can act as a Lewis basic site to binding with the Li moieties in LiPSs[38]. Besides, the tetracobalt-containing POM framework $[Co_4(H_2O)_2(PW_9O_{34})_2]^{10-}$ ($\{Co_4W_{18}\}$), which is especially reminiscent of the natural oxygen-evolving complex $Mn_4CaO_4$ in photosystem II, has been explored for photocatalytic and electrocatalytic oxygen-evolution reaction, as well as the lihtium–oxygen batteries[42–45]. However, POM clusters are usually endowed with low electrical conductivity and tend to agglomerate, which would limit the charge transfer of electrocatalytic process and increase the activation energy barrier.

Herein, we developed a single-dispersed polyoxometalate molecular cluster catalyst where the monodisperse $\{Co_4W_{18}\}$ clusters are uniformly embedded on the reduced graphene oxide (rGO) to form the $\{Co_4W_{18}\}$/rGO composites with the positively charged poly(ethyleneimine) (PEI) as the linker via a synthetic approach of electrostatic assembly. The dehydrated $\{Co_4W_{18}\}$ clusters can expose unsaturated cobalt atoms with enhanced polysulfide adsorption that would reduce the activation energy of LiPS conversion from 0.55 eV in pristine rGO to 0.3 eV in $\{Co_4W_{18}\}$/rGO. Owing to the rapid interfacial charge transfer and the exposure of abundant active sites, $\{Co_4W_{18}\}$/rGO composites could achieve more homogenized $Li_2S$ deposition and lower the activation barrier of $Li_2S$, compared with the physical mixture of $\{Co_4W_{18}\}$ and rGO ($\{Co_4W_{18}\}$+rGO). Thus, $\{Co_4W_{18}\}$/rGO composites can serve as a bifunctional electrocatalyst for the expeditious conversion between LiPSs and the insoluble $Li_2S$, which is believed to be the rate-determining step of sulfur redox reactions in Li–S batteries. When tested in full Li–S coin-cell configuration, $\{Co_4W_{18}\}$/rGO/S cathodes could exhibit a long-term cycling with a capacity fading of 0.015% per cycle after 1000 cycles at 2 C (i.e., 3.36 A g$^{-1}$). An areal capacity of 4.55 mAh cm$^{-2}$ is also achieved with a sulfur loading of 5.6 mg cm$^{-2}$ and E/S ratio of 4.5 μL mg$^{-1}$. Moreover, a Li–S single-electrode pouch cell based on the single-side coated $\{Co_4W_{18}\}$/rGO/S cathode (active area of 15 × 4 cm$^2$) demonstrates a specific capacity of about 800 mAh g$^{-1}$ at a sulfur loading of 3.6 mg cm$^{-2}$ for 100 cycles at 0.2 C (i.e., 336 mA g$^{-1}$) with E/S ratio of 5 μL mg$^{-1}$.

## Results and discussion

**Preparation and characterization of $\{Co_4W_{18}\}$/rGO.** The synthetic route of uniformly dispersing the $\{Co_4W_{18}\}$ clusters on the reduced graphene oxide is illustrated in Fig. 1a. At the first step, the water-soluble positively charged poly(ethyleneimine) (PEI) was introduced and adsorbed on the graphene-oxide (GO) sheets (Fig. 1b), which could function as a stabilizer to greatly suppress the undesired agglomeration during its chemical reduction process, resulting in well-dispersed reduced GO sheets (rGO)[46]. The as-obtained PEI-stabilized rGO can sequentially assemble with $\{Co_4W_{18}\}$ polyanion clusters through electrostatic interaction, which could be demonstrated by the more negative zeta potential of $\{Co_4W_{18}\}$/rGO (−24.5 mV) than that of PEI-stabilized rGO (−19.8 mV) (Fig. 1c and Fig. 1d).

As shown in Fig. 1e, the scanning electron microscopy (SEM) image of $\{Co_4W_{18}\}$/rGO reveals the typical morphology of abundant wrinkles that are similar with rGO, while the agglomerates could be obviously observed in the $\{Co_4W_{18}\}$+rGO samples due to the unachievable assembly process during the physical mixing of $\{Co_4W_{18}\}$ and rGO sheets (Supplementary Fig. 1). The elemental analysis also manifests that the mass content of cobalt element in the $\{Co_4W_{18}\}$+rGO is lower than the

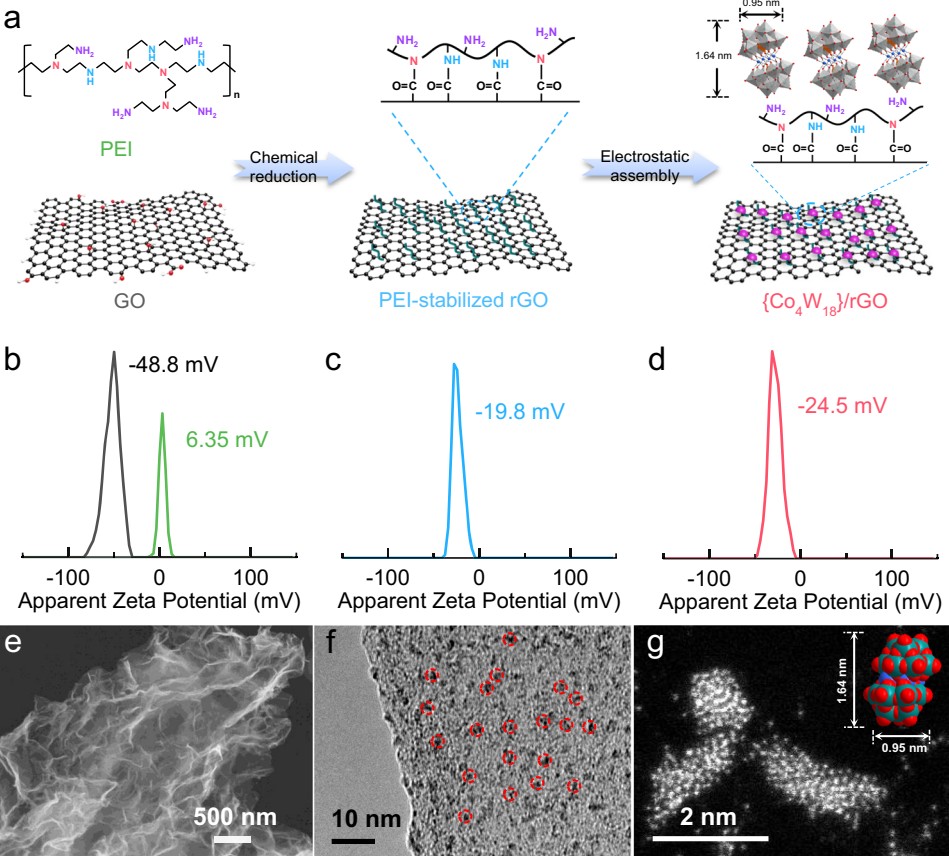

**Fig. 1 Preparation and characterization of {Co₄W₁₈}/rGO. a** Schematic diagram of the synthesis of {Co₄W₁₈}/rGO materials. **b–d** Apparent zeta potentials of GO, PEI, PEI-stabilized rGO and {Co₄W₁₈}/rGO. **e** SEM image, **f** HRTEM image, and **g** HAADF-STEM image of {Co₄W₁₈}/rGO sample.

one in the {Co₄W₁₈}/rGO even though they have the similar tungsten content (Supplementary Fig. 2). This means that the assembly synthetic strategy facilitates the exposure of cobalt to the surface of the composites. As displayed in the high-resolution transmission electron microscopy (HRTEM) images of {Co₄W₁₈}/rGO (Fig. 1f and Supplementary Fig. 3), plentiful spotted nanoparticles are evenly dispersed around the composites, which is completely unobservable in the rGO samples. This also can be supported by the homogeneous distribution of elements (Supplementary Fig. 4). The high-angle annular dark-field scanning transmission electron microscopy (HAADF-STEM) was conducted to further disclose the structure of the nanoparticles on the atomic scale (Fig. 1g). Definitely, the corresponding nanoparticles with the size about 2 nm match the individual {Co₄W₁₈} cluster with the length of 1.64 nm and the width of 0.95 nm, thus confirming the achievement of monodisperse {Co₄W₁₈} clusters throughout the rGO matrix.

In addition, the {Co₄W₁₈} clusters maintain high thermal stability where the infrared characteristic absorption peaks of P–O (1037 cm⁻¹), W = O (939 cm⁻¹), W–O–W (882 cm⁻¹ and 791 cm⁻¹), and Co–O (721 cm⁻¹) bond could still be remained after being dehydrated at 280 °C for 10 h under the Ar atmosphere. This is further confirmed by the comparison of powder X-ray diffraction (XRD) patterns of pristine {Co₄W₁₈} samples, dehydrated and rehydrated samples (Supplementary Fig. 5). After being loaded on the rGO sheets, the infrared characteristic absorption peaks of {Co₄W₁₈} clusters still keep unchanged, indicating its structural stability in the {Co₄W₁₈}/rGO composites (Supplementary Fig. 6). The pore structures of materials were investigated by the N₂ adsorption-desorption analysis (Supplementary Fig. 7). The {Co₄W₁₈}/rGO composites

inherit well the original porous structure with relatively concentrated pore size at the range of 2–5 nm, but exhibit a slight reduction in the specific surface area (97.77 m² g⁻¹), compared with the initial rGO (107.52 m² g⁻¹). The large surface area and plenty of mesoporous structure would be conducive to accommodate the active sulfur and physically confine the lithium polysulfides (LiPSs) for shuttle inhibition. Besides, Raman experiments reveal that rGO and {Co₄W₁₈}/rGO composites show the approximate intensity ratios of D band and G band, suggesting their similar graphitization (Supplementary Fig. 8).

**Chemical adsorption interaction of polysulfides**. The surface adsorption of LiPSs is the primary condition for heterogeneous electrocatalysis of LiPS conversion reactions, and the adsorption ability always has been taken to determine the performance of related host materials for Li-S batteries. According to the molecular structure of the [Co₄(H₂O)₂(PW₉O₃₄)₂]¹⁰⁻ polyanion cluster (Supplementary Fig. 5), the coordinated water would be removed after dehydration, thus leaving the two cobalt atoms at the edge of the cluster in unsaturated state. The unsaturated cobalt atoms afford much more Lewis acidic sites to bind with the polysulfide moieties, as well as the terminal oxygen atoms ligating to Li cations. The enhanced LiPS adsorption of {Co₄W₁₈}/rGO composites can be visualized by immersing different materials into the Li₂S₆ solution as presented in the inset of Fig. 2a. Notably, the color of the Li₂S₆ solution changed from yellow to nearly transparent after adsorption by {Co₄W₁₈}/rGO composites for 3 h. By contrast, the Li₂S₆ solution with the rGO sample still remained yellow clearly, indicating that {Co₄W₁₈} clusters could enhance the adsorption capacity of polysulfides. This can be also

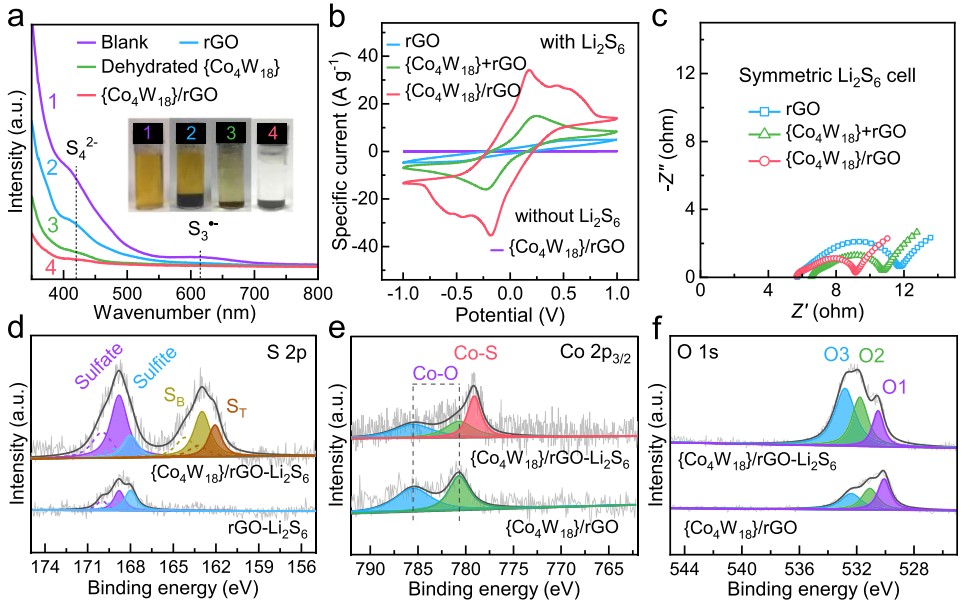

**Fig. 2 Adsorption and conversion reactions of polysulfides. a** UV–vis adsorption spectra of $Li_2S_6$ solution before and after adsorption by various materials for 3 h (inset figure: optical images of the above solutions). **b** CV curves and **c** Nyquist plots of $Li_2S_6$ symmetrical cells with the rGO, $\{Co_4W_{18}\}$+rGO, and $\{Co_4W_{18}\}$/rGO electrodes. XPS profiles of **d** S 2p, **e** Co 2p, and **f** O 1s peaks for rGO and $\{Co_4W_{18}\}$/rGO before and after adsorption of polysulfides.

demonstrated by the pale-yellow $Li_2S_6$ solution with the individual dehydrated $\{Co_4W_{18}\}$ sample, despite that the overall amount of accessible adsorption sites are limited owing to the agglomeration. Moreover, as shown in Fig. 2a, the UV–vis absorption spectrum of the pristine $Li_2S_6$ solution exhibits two characteristic peaks at around 420 nm and 615 nm, which can be attributed to $S_4^{2-}$ and $S_3^{\cdot-}$ species, respectively[47–50]. This inconsistency can be well explained that the $S_4^{2-}$ species have a high negative charge density, which could be better thermodynamically stabilized by weakly solvated $Li^+$ prevalent in the conventional solvent mixture of 1,3-dioxolane and 1,2-dimethoxyethane (DOL/DME) via Pearson's hard–soft acid–base theory[51]. These peaks would be subjected to the largest intensity decline after adsorption by $\{Co_4W_{18}\}$/rGO composites, implying the lowest LiPS concentration in the supernatant and confirming its strongest LiPS adsorption capability.

Symmetric cells were assembled based on the sulfur-free electrodes and $Li_2S_6$ electrolyte to explore the reaction kinetics of polysulfide conversion. Compared with rGO and $\{Co_4W_{18}\}$+rGO, the cyclic voltammetry (CV) curve of $\{Co_4W_{18}\}$/rGO symmetric cell exhibits the highest current response, suggesting its rapider and more sufficient LiPS conversion on the electrode surface (Fig. 2b). Meanwhile, the CV profiles of $\{Co_4W_{18}\}$/rGO symmetric cell display good coincidence for the first 5 cycles (Supplementary Fig. 9), signifying the good reversibility of LiPS redox reactions. As displayed in Fig. 2c and Supplementary Fig. 10, the corresponding electrochemical impedance spectra (EIS) were measured with three identical cells at the same conditions. The $\{Co_4W_{18}\}$+rGO and $\{Co_4W_{18}\}$/rGO electrodes exhibit a visibly lower charge-transfer resistance in comparison with rGO electrodes, further confirming the catalytic effect of $\{Co_4W_{18}\}$ clusters for polysulfide conversion. Furthermore, the ohmic resistance of $\{Co_4W_{18}\}$+rGO would be slightly higher than that of rGO electrode due to the introduction of $\{Co_4W_{18}\}$ semiconductor materials, while $\{Co_4W_{18}\}$/rGO composites well maintain the electrical conductivity of original rGO according to the nearly equivalent ohmic resistance.

X-ray photoelectron spectroscopy (XPS) was carried out to elucidate the chemical interaction between the composites and

LiPSs. In the XPS spectra of S 2p (Fig. 2d), only two weak peaks could be observed at the high binding-energy region between 166 eV and 172 eV, corresponding to the sulfite and sulfate formed on the surface of rGO after interacting with LiPSs[52]. In contrast, $\{Co_4W_{18}\}$/rGO composites could demonstrate stronger intensity response and two additional peaks located at 161.9 and 163.1 eV, which can be assigned to the terminal sulfur ($S_T$) and bridging sulfur ($S_B$) of LiPSs, respectively[53]. This further confirms the stronger chemical adsorption of $\{Co_4W_{18}\}$/rGO composites for LiPSs. Moreover, as shown in Fig. 2e, the XPS spectra of Co $2p_{3/2}$ for $\{Co_4W_{18}\}$/rGO can be separated into two characteristic peaks located at 780.7 and 785.6 eV, which are ascribed to the Co–O bonds in the $\{Co_4W_{18}\}$ cluster. After adsorbing LiPSs, an additional peak at 778.9 eV could be observed, demonstrating the formation of a Co–S bond[54]. In the XPS spectra of O 1s (Fig. 2f), three deconvoluted peaks (O1, O2, and O3) could be differentiated. The O1 peak located at 530.1 eV indicates the lattice oxygen, such as W–O, W=O and Co–O bonds. The O2 peak emerging at 531.1 eV is associated with the surface binding oxygen, such as the hydroxyl and carboxyl. The O3 peak centered at 532.4 eV can be assigned to the adsorbed oxygen[55]. After adsorption with LiPSs, all the O 1s peaks shift toward higher binding energy with the enhanced intensity corresponding to the reduced electron cloud density of oxygen, indicating the interaction between polysulfides and all oxygenic species on the surface of $\{Co_4W_{18}\}$/rGO. Among them, the lattice oxygen (O1 peak) and the surface binding oxygen (O2 peak) can act as Lewis bases, which could show the "lithium bond"-like interaction with polysulfides, while the adsorbed oxygen (O3 peak) would play an important role for the formation of sulfite and sulfate species mentioned above. These results evidentially demonstrate that the unsaturated cobalt atoms and terminal oxo-ligands of $\{Co_4W_{18}\}$ clusters are highly favorable for LiPS adsorption, which would offer superior LiPSs chemical confinement toward efficient shuttle inhibition.

**Bifunctional catalytic effect for $Li_2S$ deposition and oxidation.** To further investigate the catalytic effect of $\{Co_4W_{18}\}$ for sulfur redox reactions, potentiostatic nucleation and oxidation of $Li_2S$

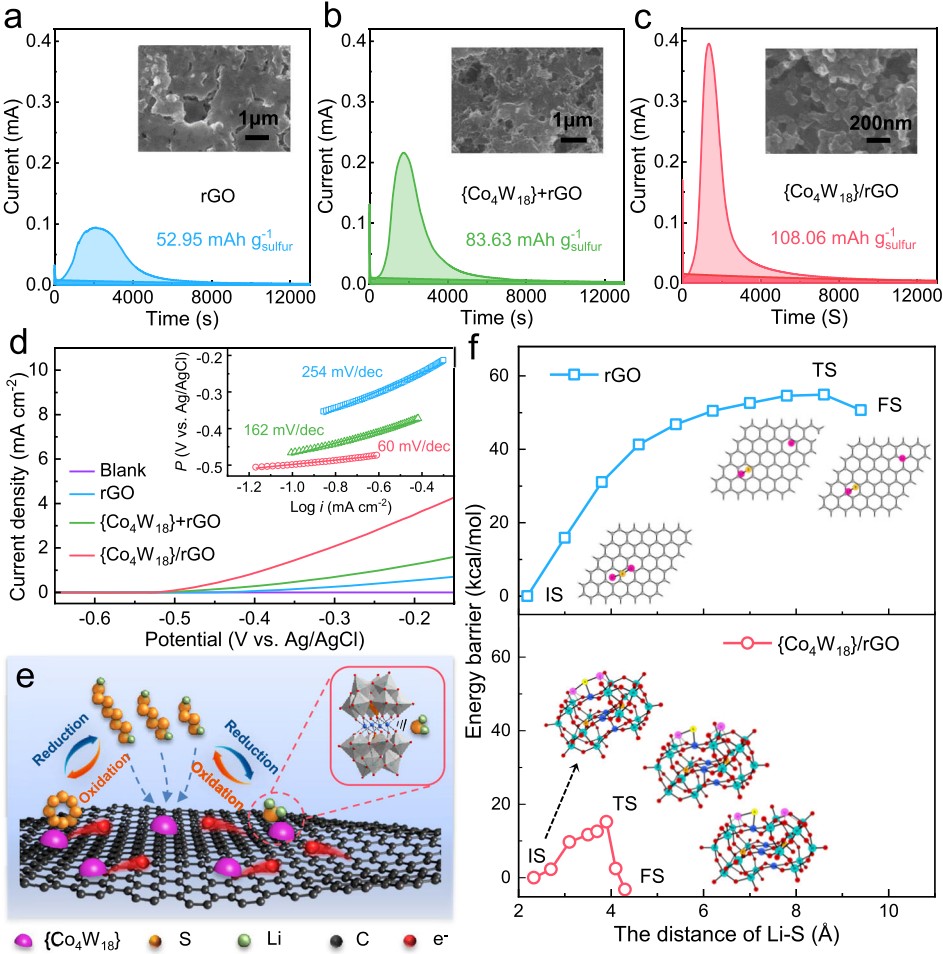

**Fig. 3 Bifunctional catalytic behaviors of Li$_2$S deposition and oxidation. a–c** Potentiostatic nucleation profiles of Li$_2$S$_8$ solution on the rGO, {Co$_4$W$_{18}$} +rGO, and {Co$_4$W$_{18}$}/rGO electrodes. The dark and light areas correspond to the reduction of Li$_2$S$_8$/Li$_2$S$_6$ and the Li$_2$S deposition, respectively. The insets are the SEM images of electrodes after Li$_2$S deposition. **d** LSV curves and Tafel plots of Li$_2$S oxidation on the rGO, {Co$_4$W$_{18}$}+rGO, and {Co$_4$W$_{18}$}/rGO electrodes. **e** Schematic illustration of bifunctional catalytic effect for Li$_2$S deposition and oxidation on the surface of {Co$_4$W$_{18}$}/rGO composites. **f** Energy profiles for Li$_2$S → LiS + Li on rGO (blue line) and {Co$_4$W$_{18}$} (red line), respectively. The inserted pictures are the optimized structures of the initial state (IS), transition state (TS), and final state (FS).

on the rGO, {Co$_4$W$_{18}$}+rGO and {Co$_4$W$_{18}$}/rGO electrodes were conducted to clarify the involved multiphase transition process[56]. As shown in the potentiostatic nucleation profiles (Fig. 3a–c and Supplementary Fig. 11), the capacity of precipitated Li$_2$S on {Co$_4$W$_{18}$}/rGO (108.06 mAh g$_s^{-1}$) is higher than that on rGO (52.95 mAh g$_s^{-1}$) and {Co$_4$W$_{18}$}+rGO (83.63 mAh g$_s^{-1}$). Besides, the higher current intensity indicates the fast LiPS trapping and deposition of Li$_2$S on the {Co$_4$W$_{18}$}/rGO electrode. The deposited morphology of Li$_2$S was characterized via SEM images. As presented in the insets, the discharge products are uniformly distributed on the {Co$_4$W$_{18}$}/rGO cathode, while the blocky Li$_2$S aggregation would be formed on the rGO and {Co$_4$W$_{18}$}+rGO cathode, which also are demonstrated by the corresponding elemental mapping (Supplementary Fig. 12 and Fig. 13). The differences of deposition behaviors fully reveal the strong chemical adsorption of LiPSs on the {Co$_4$W$_{18}$}/rGO cathode with the abundant active sites, achieving its superior catalytic activity for LiPSs conversion and Li$_2$S deposition[57,58]. In addition, the {Co$_4$W$_{18}$}/rGO electrode exhibits stronger peak intensity of Li$_2$S products from XRD patterns (Supplementary Fig. 14). The clean and smooth appearance of corresponding lithium anode and separator further confirms the enhanced LiPS absorption capability of {Co$_4$W$_{18}$}/rGO for increased retention of active materials and mitigated lithium corrosion.

Three-electrode linear-sweep voltammetry (LSV) measurements were then conducted to investigate the catalytic oxidation behaviors of Li$_2$S on the surface of these electrodes (Fig. 3d). {Co$_4$W$_{18}$}+rGO and {Co$_4$W$_{18}$}/rGO electrodes exhibit the lower onset potential of −0.516 V and −0.525 V for Li$_2$S oxidation compared with the rGO electrode (−0.445 V), indicating the considerably reduced energy barrier and enhanced reaction kinetics with the presence of {Co$_4$W$_{18}$} clusters. This result can be further supported by the Tafel plots of Li$_2$S oxidation reaction as presented in the inset of Fig. 3d. The minimal Tafel slope of 60 mV dec$^{-1}$ could be achieved for {Co$_4$W$_{18}$}/rGO electrode, comparing with the {Co$_4$W$_{18}$}+rGO (162 mV dec$^{-1}$) and rGO electrodes (254 mV dec$^{-1}$), which are ascribed to the synergistic effect between {Co$_4$W$_{18}$} and rGO. On the one hand, the hierarchical mesoporous architecture and high electrical conductivity afford a large specific surface area and ample active interfaces. On the other hand, the enriched unsaturated cobalt atoms in the {Co$_4$W$_{18}$}/rGO matrixes endow abundant absorption sites as well as catalytic sites to facilitate the Li$_2$S oxidation. Thus, the introduction of {Co$_4$W$_{18}$} provides bifunctional catalytic effect for Li$_2$S deposition and oxidation (Fig. 3e).

Density-functional theory (DFT) calculations were performed to well understand the promoting effect of {Co$_4$W$_{18}$} on Li$_2$S decomposition (Fig. 3f). Structurally, Li$_2$S would stride on

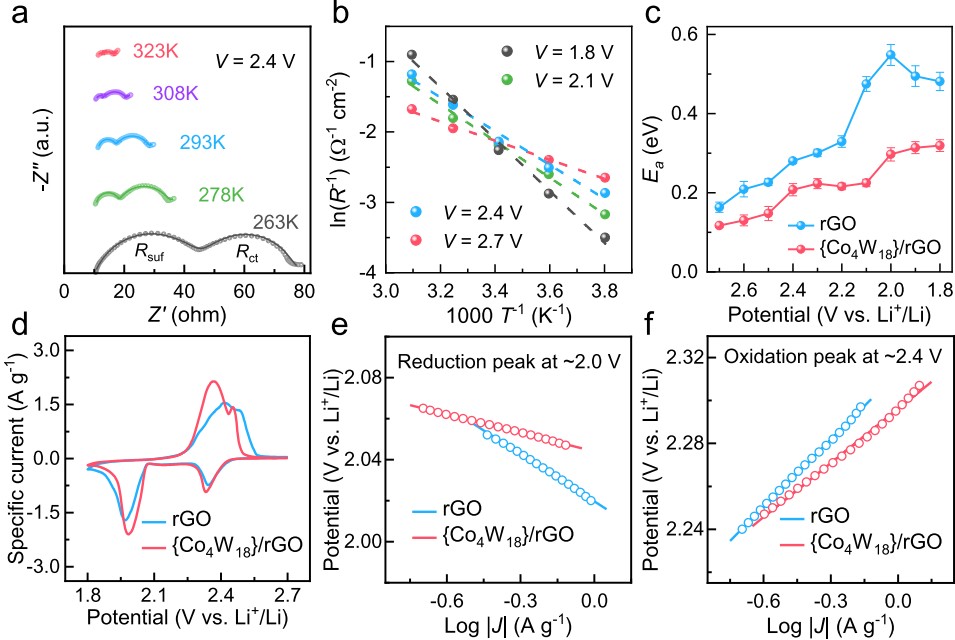

**Fig. 4 Activation-energy profiles and kinetic analyses of sulfur redox reactions. a** Nyquist plots of {Co₄W₁₈}/rGO/S cathodes under different temperatures at 2.4 V. **b** Arrhenius plots showing the proportional relation between logarithmic values of the reciprocal of charge-transfer resistance and the reciprocal of the absolute temperatures for 2.7 V, 2.4 V, 2.1 V, and 1.8 V. **c** Activation-energy profiles for rGO/S and {Co₄W₁₈}/rGO/S cathodes at various potentials. Error bars derived from the standard deviation of three independent electrodes and the fitting error. **d** The fifth-cycle CV profiles of the rGO/S and {Co₄W₁₈}/rGO/S cathodes at a scan rate of 0.1 mV s⁻¹. **e, f** Tafel plots calculated from the reduction peak at ~2.0 V and oxidation peak at ~2.4 V.

{Co₄W₁₈} with its S atom bonding with unsaturated–coordinated Co atoms (Lewis acid sites), while Li atoms interacting with terminal or bridging O atoms (Lewis base sites). As a result, one of the Li–S bonds would be remarkably activated with the bond distance of Li–S elongating from 2.190 Å to 2.329 Å. It is expected that {Co₄W₁₈} would stabilize the decomposition state of Li₂S and the corresponding transition state (TS) as well. Indeed, the results show that Li₂S decomposition on {Co₄W₁₈} only encounters a small barrier of 15.2 kcal/mol. By contrast, the decomposition on graphene surface would be a strongly endothermic process with a barrier of 54.9 kcal/mol. These findings nicely explain why introducing {Co₄W₁₈} into the lithium–sulfur battery would enhance the electrocatalytic performance for Li₂S oxidation.

**Activation energy and kinetics of sulfur redox reactions.** To validate the electrocatalytic reactivity of {Co₄W₁₈}/rGO on practical Li–S coin cells, sulfur composites with sulfur content of approximately 70 % were prepared via melt-impregnation process (Supplementary Fig. 15). First, the activation energy ($E_a$) for each step of LiPS conversion process could be determined by detecting the charge transfer resistances of sulfur-composite cathodes at the corresponding potentials under different temperatures[59]. Thus, EIS measurements were performed from 2.7 V to 1.8 V under different temperature (263, 278, 293, 308 or 323 K) in a frequency range from 0.1 Hz to 100 kHz. The equivalent circuit presented in Supplementary Fig. 16 would be used to fit the impedance spectra, where the first semicircle is assigned to Li-ion diffusion through the surface nanolayer ($R_{suf}$), the second semicircle is ascribed to charge-transfer resistance ($R_{ct}$), and the tail line indicates the Warburg resistance ($Z_w$)[60]. As shown in Supplementary Figs. 17 and 18, the raw data and fitted data were displayed as symbols and lines, respectively, with good fitting effect. According to the Arrhenius equation, the logarithmic values of the reciprocal of charge-transfer resistance depend linearly on the reciprocal of the absolute temperatures.

By fitting the charge transfer resistance measured at different temperatures (Fig. 4a), the $E_a$ at the corresponding potentials could be figured out with the slope of the linear fitting (Fig. 4b). As resulted in Fig. 4c, the activation energies are reduced comparatively with the introduction of {Co₄W₁₈}, especially at 2.0–1.8 V. The maximum value of $E_a$ would decrease from 0.55 eV in rGO to 0.3 eV in {Co₄W₁₈}/rGO, again confirming that {Co₄W₁₈} could facilitate the conversion of LiPSs into insoluble Li₂S₂/Li₂S products.

The cyclic voltammetry (CV) measurements were also carried out. As shown in Fig. 4d, the fifth-cycle CV profiles of sulfur-composite cathodes exhibit two conventional reduction peaks at the potential of around 2.35 V and 2.0 V, representing the multiphase changing from the solid S₈ to soluble LiPSs, then to insoluble Li₂S₂/Li₂S. In the reverse scanning, two corresponding oxidation peaks at 2.3–2.5 V can be put down to the activation of Li₂S into LiPSs, and finally to initial S₈. Comparing with rGO/S, the asymmetric cell with {Co₄W₁₈}/rGO/S cathode presents higher redox-peak current and lower polarization for the deposition and oxidation of Li₂S, which are in line with the results in Fig. 3. Notably, the depressed polarization could be observed after the first cycle of sulfur redox reactions (Supplementary Fig. 19), which should arise from the dissolution of polysulfides and the redistribution of active sulfur with more access to electrons and ions. In the succeeding cycles, the CV profiles of {Co₄W₁₈}/rGO/S cathode well maintain, suggesting the good reversibility of sulfur conversion. To explicitly show the catalytic function of {Co₄W₁₈}, Tafel plots of the reduction (from Li₂Sₙ to Li₂S at ~2.0 V) and oxidation (from Li₂S to Li₂Sₙ at ~2.4 V) peaks with rGO/S and {Co₄W₁₈}/rGO/S cathodes are presented in Fig. 4e, f. For the reduction process, the fitted Tafel slopes of the rGO/S and {Co₄W₁₈}/rGO/S cathodes are 76.65 and 30.1 mV dec⁻¹, respectively. The slopes are 104.81 and 84.04 mV dec⁻¹ in the oxidation process. Compared with rGO/S, the lower slopes of reduction and oxidation process with the {Co₄W₁₈}/

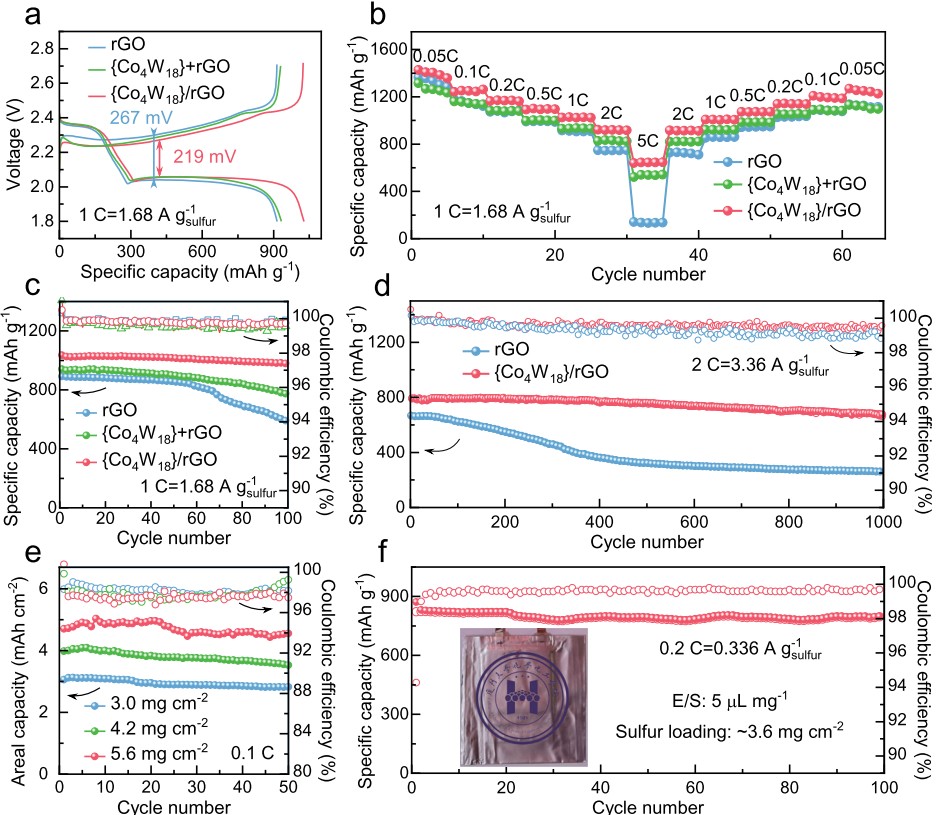

**Fig. 5 Electrochemical performance of Li–S cells based on various sulfur cathodes. a** Galvanostatic discharge–charge curves at 1 C (i.e., 1.68 A $g_s^{-1}$). **b** Rate performance of various sulfur cathodes at different specific current. **c** Cycle performance at 1 C (i.e., 1.68 A $g_s^{-1}$). **d** Long-term cycling stability at 2 C (i.e., 3.36 A $g_s^{-1}$). **e** Cycle performance of {Co$_4$W$_{18}$}/rGO/S cathodes with high sulfur loading at 0.1 C (i.e., 0.168 A $g_s^{-1}$). **f** Cycle performance of the pouch cell with sulfur loading of 3.6 mg cm$^{-2}$ and E/S ratio of 5 μL mg$^{-1}$ at 0.2 C (i.e., 0.336 A $g_s^{-1}$).

rGO/S cathode clearly demonstrate its bifunctional catalytic effect for the rapid conversion between LiPSs and Li$_2$S.

**Electrochemical energy-storage performances of the bifunctional electrocatalyst in Li–S cells.** The electrochemical performance of various sulfur cathodes at different specific current was tested in coin-cell configuration. As illustrated in Fig. 5a and Supplementary Fig. 20, the galvanostatic discharge–charge curves of various sulfur-composite electrodes were compared at different rates of 0.1 C, 1 C, and 2 C (corresponding to the specific current of 0.168, 1.68, and 3.36 A $g_s^{-1}$). The voltage difference of discharging and charging shows little change at the relatively low rate of 0.1 C for the rGO/S (121 mV) and {Co$_4$W$_{18}$}/rGO/S cathodes (113 mV). But at the higher rate of 1 C and 2 C (i.e., 1.68 and 3.36 A $g_s^{-1}$), the reaction polarization for rGO/S cathode remarkably increases to 267 mV and 442 mV, while the {Co$_4$W$_{18}$}/rGO/S cathode performs modestly raised overpotentials of 219 mV and 295 mV. Apart from that, the activation barrier of Li$_2$S could be reduced on the {Co$_4$W$_{18}$}/rGO electrode in comparison with rGO electrode (Supplementary Fig. 20), demonstrating a further evidence of the more prominent reaction kinetics for Li$_2$S oxidation on the {Co$_4$W$_{18}$}/rGO cathode. Moreover, the rate capabilities in Fig. 5b demonstrate that the {Co$_4$W$_{18}$}/rGO/S cathode exhibits the best performance, which delivers the considerable reversible specific capacity of 1426 mAh g$^{-1}$, 1251 mAh g$^{-1}$, 1171 mAh g$^{-1}$, 1100 mAh g$^{-1}$, 1028 mAh g$^{-1}$, 919 mAh g$^{-1}$, and 644 mAh g$^{-1}$ at different rates of 0.05 C, 0.1 C, 0.2 C, 0.5 C, 1 C, 2 C, and 5 C (i.e., 0.084, 0.168, 0.336, 0.84, 1.68, 3.36, and 8.4 A $g_s^{-1}$), respectively (Supplementary Fig. 21).

As presented in Fig. 5c, good cycling stability could be achieved for the {Co$_4$W$_{18}$}/rGO/S cathode. After 100 cycles at 1 C (i.e.,

1.68 A $g_s^{-1}$), the specific discharge capacity can still maintain above 980 mAh g$^{-1}$, and the coulombic efficiency of each cycle is nearly 100%. In contrast, rGO/S and {Co$_4$W$_{18}$}+rGO/S cathodes show an evident capacity fading. With the introduction of {Co$_4$W$_{18}$} clusters, the initial discharge capacity of sulfur composites could be increased from 890 mAh g$^{-1}$ to 1030 mAh g$^{-1}$ and the capacity fading is distinctly mitigated, explicitly suggesting that {Co$_4$W$_{18}$} clusters are beneficial for high sulfur utilization and reduced dissolution loss of polysulfides. Remarkably, {Co$_4$W$_{18}$}/rGO/S cathode could still release a specific capacity of around 670 mAh g$^{-1}$ after 1000 cycles at the relatively higher rate of 2 C (i.e., 3.36 A $g_s^{-1}$) with a capacity decay of 0.015 % per cycle, while the specific capacity of rGO/S cathode is progressively declined to only 260 mAh g$^{-1}$ after the long-term cycles (Fig. 5d and Supplementary Table 1). Moreover, the surface morphology of the lithium electrodes after cycling at different rates (1 C, 2 C and 5 C, i.e., 1.68, 3.36 and 8.4 A $g_s^{-1}$) has been investigated (Supplementary Fig. 22). The deposition of metallic lithium would become rough with the increasing of specific current. Many black spots appear on the surface of the lithium-metal electrode after cycling for 500 cycles at 2 C (i.e., 3.36 A $g_s^{-1}$), indicating the formation of "dead lithium" (i.e., parts of the lithium metal electrode which are electronically disconnected from the current collector), which has been believed to be one of the most important reasons for the decline of coulombic efficiency[61]. Furthermore, it has been proved that the {Co$_4$W$_{18}$}/rGO host material itself has no capacity contribution during the discharging and charging process of Li–S cells. The cell was assembled with the sulfur-free {Co$_4$W$_{18}$}/rGO material as the cathode, lithium metal as the anode, and identical Li–S electrolyte. Then galvanostatic charge–discharge tests and cyclic

voltammetry were also conducted in the voltage range of 1.8–2.7 V (Supplementary Fig. 23). Almost negligible capacity ($\sim$1 mAh g$^{-1}$) and the absence of any redox peaks indicates that $\{Co_4W_{18}\}$/rGO material would not undergo any electrochemical reactions with high structural stability.

In the end, to meet the requirement of high-energy density for the practical application of Li–S cells, the $\{Co_4W_{18}\}$/rGO/S composites were explored in coin cell configuration at a high sulfur loading of 3.0 mg cm$^{-2}$, 4.2 mg cm$^{-2}$ and 5.6 mg cm$^{-2}$ with an invariable amount of electrolyte (50 µL), respectively. As shown in Fig. 5e, $\{Co_4W_{18}\}$/rGO/S cathode with high sulfur loading can still keep good cycling stability. Particularly, when the sulfur loading is increased to 5.6 mg cm$^{-2}$ (the ratio of electrolyte to sulfur, E/S = 4.5 µL mg$^{-1}$), the Li–S coin cell could exhibit a high areal capacity of 4.55 mAh cm$^{-2}$ with a coulombic efficiency of above 98% per cycle after 50 cycles at 0.1 C (i.e., 0.168 A g$_s^{-1}$), which is competitive in comparison with the state-of-the-art single-atom host materials at such low E/S ratio (Supplementary Table 2). As a demonstration, a single-electrode pouch cell based on the winding single-side-coated $\{Co_4W_{18}\}$/rGO/S cathode (active area of 15 × 4 cm$^2$) was fabricated with a sulfur loading of 3.6 mg cm$^{-2}$ and an E/S ratio of 5 µL mg$^{-1}$. As shown in Fig. 5f, the pouch cell delivers a reversible capacity of 795 mAh g$^{-1}$ after 100 cycles at 0.2 C (i.e., 0.336 A g$_s^{-1}$), meaning that a high output capacity of 172 mAh can be achieved. Besides, the light-emitting diode (LED) lamps could be powered normally by the assembled cell, even under a 90-degree bend and a 180-degree fold (Supplementary Fig. 24), indicating that $\{Co_4W_{18}\}$/rGO also shows great potential for flexible energy-storage systems.

In summary, we successfully develop single-dispersed $\{Co_4W_{18}\}$ clusters on the surface of rGO with the positively charged poly(ethyleneimine) as a linker. The dehydrated $\{Co_4W_{18}\}$ clusters could expose unsaturated cobalt atoms with enhanced polysulfide adsorption, which would reduce the activation energy of LiPSs conversion from 0.55 eV in pristine rGO to 0.3 eV in $\{Co_4W_{18}\}$/rGO. Moreover, $\{Co_4W_{18}\}$/rGO composites could achieve more homogenized $Li_2S$ deposition and lower the activation barrier of $Li_2S$ due to the exposure of abundant active sites, compared with $\{Co_4W_{18}\}$+rGO. Thus, $\{Co_4W_{18}\}$/rGO composites present the bifunctional catalytic effect for the expeditious conversion between LiPSs and the insoluble $Li_2S$. When applied for sulfur host materials, $\{Co_4W_{18}\}$/rGO/S cathodes could show remarkable rate performance and cycling stability which still deliver a specific capacity of about 670 mAh g$^{-1}$ with a capacity fading of 0.015 % per cycle after 1000 cycles at 2 C (i.e., 3.36 A g$_s^{-1}$). In addition, the cathodes exhibit a high areal capacity of 4.55 mAh cm$^{-2}$ at high sulfur loading (5.6 mg cm$^{-2}$) and poor electrolyte (E/S ratio: 4.5 µL mg$^{-1}$), demonstrating its great potential for practical application. This work illustrates the catalytic mechanism of the single-polyoxometalate molecular cluster catalysts for sulfur redox reactions from an atom-level view. The essential interactions between catalysts and sulfur species have been indicated and clarified. Moreover, thanks to the structural and chemical tunability of POM clusters, the structure–property–function relationships can be established, which would provide a promising strategy to design advanced catalysts for other catalytic fields beyond sulfur conversion reactions.

## Methods

**Reagents and materials**. All chemical reagents were purchased from Aladdin (Shanghai, China), and were of analytic grade (AR) without any purification before use. Ultrapure water (18.2 MΩ·cm$^{-1}$ resistivity) was used in all experiments. The Toray carbon paper (TGP-H-060, thickness of 0.19 mm) was applied from Sinero Technology Co., Ltd (Suzhou, China). The Li-metal foils ($\Phi$15.8 × 1.0 mm, ≥99.9%) were purchased from Shunyou Metal Co., Ltd. (Shanghai, China). About 0.5 M lithium trifluoromethanesulfonate (LiCF$_3$SO$_3$) + 0.5 M lithium nitrate (LiNO$_3$) dissolved in the mixed solvent of 1,3-dioxolane (DOL, H$_2$O < 20 ppm) and 1,2-

dimethoxyethane (DME, H$_2$O < 20 ppm) (v/v, 1:1) were purchased from Fosai New Materials Co. Ltd (Suzhou, China).

**Synthesis of poly(ethyleneimine)-stabilized rGO**. Graphene-oxide (GO) sheets were pre-prepared by oxidation and exfoliation of expansible graphite according to the modified Hummers method[62]. The GO nanosheets were then redispersed in deionized water to obtain a uniform dispersion (2 mg/mL GO). About 100 mL of 2 mg/mL poly(ethyleneimine) (PEI) aqueous solution was added slowly into the above GO dispersion (100 mL), followed by vigorously stirring for 12 h at 60 °C. Then the mixed solution was treated with 200 µL of hydrazine hydrate (85%), and was refluxed at 95 °C for 12 h under N$_2$ atmosphere. The PEI-stabilized rGO samples can be fabricated by washing with deionized water for several times and subsequent freeze-drying.

**Synthesis of $\{Co_4(PW_9O_{34})_2\}$ cluster-modified rGO ($\{Co_4W_{18}\}$/rGO)**. K$_{10}$[Co$_4$(H$_2$O)$_2$(PW$_9$O$_{34}$)$_2$]·22H$_2$O was prepared following the method reported by Evans and Weakley in 1986[63]. About 110 g (0.33 mol) of Na$_2$WO$_4$·2H$_2$O and 7.5 g (0.042 mol) of Na$_2$HPO$_4$·2H$_2$O was dissolved in 350 mL of water. The pH was adjusted to 7.0 with CH$_3$COOH. The mixed solution was heated to boiling with magnetic stirring, into which 20.8 g (0.084 mol) Co(CH$_3$COO)$_2$·4H$_2$O in 100 mL of water was added slowly, and then reflux continued for 2 h. The resulted deep-red solution was then treated with a hot solution of K(CH$_3$COO) (20.8 g, pH = 7) and cooled. The crude crystalline product was filtered off and washed with aqueous KNO$_3$ followed by 50% EtOH. The residue was twice reprecipitated from hot KNO$_3$ solution (85 °C), and twice recrystallized from hot water.

The above-prepared rGO was redispersed in 100 mL of deionized water, followed by adding 50 mg K$_{10}$[Co$_4$(H$_2$O)$_2$(PW$_9$O$_{34}$)$_2$]·22H$_2$O. After continuously stirring for 12 h at room temperature, the composites were washed with deionized water for several times and subsequently freeze-dried. The $\{Co_4(PW_9O_{34})_2\}$ cluster-modified rGO ($\{Co_4W_{18}\}$/rGO) samples can be obtained by heat treatment at 280 °C for 10 h under Ar atmosphere to remove the crystal water and coordinated water. Element analysis values (wt%): C 73.42, O 9.43, N 2.41, Co 2.32, W 12.34, P 0.03, and K 0.05, corresponding to the mass loading of the $\{Co_4W_{18}\}$ and PEI, are 17.5 wt% and 7.57 wt%, respectively. As control samples, the mixture of $\{Co_4W_{18}\}$ and rGO (denoted as $\{Co_4W_{18}\}$+rGO) was fabricated by physical grinding with a mass ratio of 1:5, and subsequent heat treatment at 280 °C for 10 h under Ar atmosphere.

**Synthesis of sulfur composites**. The $\{Co_4W_{18}\}$/rGO/S composites were fabricated through a melt-diffusion method. Typically, the $\{Co_4W_{18}\}$/rGO samples and sulfur powder were mixed and grinded evenly with a mass ratio of 3:7. Then the mixture was sealed in a reactor, followed by the heat treatment at 155 °C for 12 h. The control materials of rGO/S and $\{Co_4W_{18}\}$+rGO/S were also prepared by the same method.

**Material characterizations**. Zeta potentials were probed by Zetasizer (Malvern Nano-ZS & MPT-2). Thermogravimetric analysis (TGA, SQT Q600) was implemented with a heating rate of 10 °C/min under N$_2$ atmosphere. XRD patterns were recorded by the ultima-IV X-ray diffractometer (Rigaku, Japan) using Cu Kα X-ray source at 3 kW with a goniometer radius of 285 mm. FT-IR measurements were performed on a Nicolet iS50 FTIR spectrometer (Thermo Scientific) equipped with an MCT detector with a resolution of 4 cm$^{-1}$. FT-IR spectra were recorded in the transmission mode, ranging from 400 to 4000 cm$^{-1}$ with 32 accumulative scans. The Raman spectra were recorded with an excitation wavelength of 532 nm on a XploRA confocal Raman microscope (Jobin Yvon-Horiba, France) to determine the graphitization of carbon-based materials. The specific surface areas were studied using the Brunauer–Emmett–Teller (BET) equation based on the N$_2$ adsorption–desorption measurement (ASAP2460). The pore-size distribution was calculated by the Barret–Joyner–Halenda (BJH) method. Scanning electron microscopy (SEM) and transmission electron microscopy (TEM) images were captured by Hitachi S-4800 and Tecnai F30, respectively. Atomic-resolution aberration-corrected HAADF-STEM was performed by Titan ETEM Themis. Surface-element analysis was performed by means of an energy-dispersive X-ray fluorescence spectrometer (EDX). Optical images of the samples were obtained by HUAWEI Mate 40 Pro from a fixed position under constant lighting design. XPS measurements were performed on PHI 5000 Versa Probe II (UIVAC-PHI) using monochromatic Al Kα (1486.8 eV) X-ray source at 24 W and 16 KV with a beam spot size of 100 µm.

**Li–S-cell assembly and electrochemical characterizations**. The sulfur composites, Super P and hydrophilic binder (5 wt% LA133, an aqueous dispersion of acrylonitrile copolymer), were mixed with a proportion of 7:2:1 by weight to obtain slurry, which was then coated on the carbon-paper collectors (diameter of 16 mm). After being dried in vacuum at 60 °C for 12 h, the as-prepared electrodes were used as the sulfur cathodes. Li–S cells were assembled using CR2016-type coin cells in the Ar-filled glove box (Vigor Tech, O$_2$ and H$_2$O < 1 ppm), which consisted of the sulfur cathodes, lithium foil anodes (thickness of 1 mm), and polypropylene separator (Celgard 2400). About 0.5 M LiCF$_3$SO$_3$ + 0.5 M LiNO$_3$ dissolved in the mixed solvent of 1,3-dioxolane (DOL) and 1,2-dimethoxyethane (DME) (v/v, 1:1) were

applied as the electrolyte. The amount of electrolyte injected is kept at 50 μL. The areal sulfur loading of the cathode was controlled at approximately 1.5 mg cm$^{-2}$ (E/S ratio: 16.7 μL mg$^{-1}$), while was increased to 3.0, 4.2, and 5.6 mg cm$^{-2}$ with the E/S ratio of 8.3, 5.9, and 4.5 μL mg$^{-1}$ for high-sulfur-loading cycling test. Galvanostatic charge/discharge tests for Li–S cells were conducted with the voltage ranging from 1.8 to 2.7 V (Neware, Shenzhen, China) in an environmental chamber with the constant temperature of 25 °C. Cyclic voltammetry (CV) and electrochemical impedance (EIS) measurements were carried out on a CHI660 electrochemical workstation. The CV profiles were collected at a scanning rate of 0.1 mV s$^{-1}$ from 1.8 to 2.7 V. EIS spectra were recorded in the potentiostatic mode with peak-to-peak amplitude of 5 mV, ranging the frequency from 1000 KHz to 0.1 Hz with 10 data points per decade of frequency. The EIS measurements were performed around a quasi-stationary potential.

**Polysulfide-adsorption test**. Anhydrous lithium sulfide and sulfur were mixed at a molar ratio of 1:5 and dissolved in DOL + DME (*v/v*, 1:1) solvent. A Li$_2$S$_6$ solution (5 mM) could be obtained by vigorously stirring for 24 h in the Ar-filled glove box (Vigor Tech, O$_2$ and H$_2$O < 1 ppm). About 2 mL of the as-prepared Li$_2$S$_6$ solution were taken into four glass bottles, to three of which were added 10 mg of rGO, dehydrated {Co$_4$W$_{18}$} and {Co$_4$W$_{18}$}/rGO samples, respectively. After being vigorously shaken for 1 min and placed for 3 h, these glass bottles were taken optical photos, and the supernate was filtered for UV–vis test. The materials after adsorption of polysulfides were filtered out in the glove box and were washed thoroughly using the anhydrous DME solvent to remove residual polysulfide, and then transferred into the XPS chambers by using the commercial air-isolating containers. Definitely, three different positions on the samples have been chosen to ensure the accuracy and reproducibility of XPS measurements. In data processing, all the binding energies have been referenced to the C 1 s line at 284.6 eV from adventitious carbon. XPSpeak 4.1 software was used for peak fitting. Generally, shirley is selected as the type of background, which could be optimized by changing the slope. Then appropriate peaks were added for fitting. For the divided peaks of the same element, the energy gap and area ratio should be fixed. FWHM also should be as consistent as possible. Gaussian/Lorentzian (GL) ratio would be set to the default value (80%).

**Symmetrical cell assembly and measurement**. About 0.1 M Li$_2$S$_6$ solution was prepared as the electrolyte of symmetrical cells through the same method described above. The electrodes for symmetrical cells were prepared by mixing the sulfur-free catalyst materials, Super P and LA133 binder, at a weight ratio of 5:4:1 and coating them on the carbon-paper disks (diameter of 13 mm), which were directly used as the working electrodes and counter electrodes to assemble the symmetrical cells with Celgard 2400 separator and Li$_2$S$_6$ electrolyte (75 μL). The entire assembly process was performed in the glove box (Vigor Tech, O$_2$ and H$_2$O < 1 ppm). CV and EIS measurements were carried out on a CHI660 electrochemical workstation. A least three identical cells have been tested at the same conditions. The CV profiles were collected at a scanning rate of 5 mV s$^{-1}$ from −1.0 to 1.0 V. EIS spectra were recorded with the frequency range of 1000 KHz–0.1 Hz at the open-circuit potential.

**Potentiostatic nucleation of Li$_2$S**. The 0.25 M Li$_2$S$_8$ electrolyte was prepared by reacting sulfur and anhydrous lithium sulfide at a molar ratio of 1:7 in DOL + DME (*v/v*, 1:1) solvent. The electrodes were fabricated by mixing the catalyst materials (rGO, {Co$_4$W$_{18}$}+rGO, and {Co$_4$W$_{18}$}/rGO) and LA133 binder at a weight ratio of 9:1 and coating them on the carbon-paper disks (diameter of 13 mm). The areal mass loading of catalyst materials in the cathode was controlled at approximately 0.5 mg cm$^{-2}$. For the cell assembly, 20 μL of as-made Li$_2$S$_8$ electrolyte was added onto the cathode side, and 20 μL of conventional electrolyte was injected onto the lithium foil anode side. The fabricated cells were galvanostatically discharged to 2.06 V at 0.107 mA (0.1 C), and then potentiostatically discharged at 2.05 V until the current was lower than 0.01 mA. The nucleation capacity of Li$_2$S can be figured out by the integral area of the plotted curve through Faraday's law. For further characterizing Li$_2$S deposition, cells were disassembled in the Ar-filled glove box (Vigor Tech, O$_2$ and H$_2$O < 1ppm), followed by thoroughly washing the electrodes with anhydrous DME to remove the electrolyte salts and polysulfides. Then the dried electrodes were sealed on the quartz slide with the kapton tape to avoid the air exposure for XRD measurement, while the dried samples were sealed in air-isolating containers in glove box and transferred quickly into the SEM equipment for measurements.

**Catalytic oxidation of Li$_2$S**. Three-electrode linear-sweep voltammetry (LSV) tests were performed to evaluate the catalytic activity of the samples for Li$_2$S oxidation. The preparation of working electrodes was the same as that of symmetric cells. The counter electrode and reference electrode were respectively platinum sheet (1 × 1 cm$^2$) and saturated Ag/AgCl electrode. About 0.1 M Li$_2$S dissolved in methanol solution was used as electrolyte. The LSV curves were measured using a CHI660 electrochemical workstation at a scanning rate of 5 mV s$^{-1}$ from −0.7 to −0.1 V.

**Density-functional theory calculation**. All the DFT calculations were conducted with the Gaussian 09 package using the ωB97XD functional theory. In our calculations, the W and Co atoms were treated with the SDD relativistic effective core potential. For the other atoms, 6–31 G(d) basis set was employed. For graphene

surface, we employed a cluster model, including 60 C atoms and 22 H atoms. For {Co$_4$W$_{18}$}, full-size model, i.e., [Co$_4$(PW$_9$O$_{34}$)$_2$]$^{10-}$, was used. To account for the solvent effect, the universal solvation model (PCM) was used. 1, 4-Dioxane was adopted as the solvent to simulate the electrolyte environment (DME:DOXL = 1:1) used in Li–S system. Energy profile was calculated by using constrained optimization with Li–S distances as constants. The scan calculations started from the molecular state of Li$_2$S adsorbing on {Co$_4$W$_{18}$} and graphene to their dissociated states.

**Activation energy of sulfur redox reactions**. The Li–S cells were assembled as described above and should be discharged and charged first at 1 C (i.e., 1.68 A g$_s^{-1}$) for 5 cycles before EIS tests. Then the cells with fully charged state (100% SOC) were placed into the thermostatic chamber with a specific temperature (263, 278, 293, 308, or 323 K) for 2 h. The impedance spectra would be measured at different potentials from 2.7 V to 1.8 V with the same electrode under the particular temperature. Meanwhile, three sets of EIS data were obtained from three independent electrodes, corresponding to three data values of charge transfer resistance at the same conditions. The average values with standard deviation would be used for linear fitting, and then calculating the activation energy of sulfur redox reactions at different potentials.

## Data availability
The authors declare that all the relevant data within this paper and its Supplementary Information file are available from the corresponding author upon reasonable request.

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

## Acknowledgements

We gratefully acknowledge the financial support from Innovative Research Group Project of NSFC (22021001), the NSFC projects (21975211, U1805254, U1705255, 21773192, 22072117), the National Key Research and Development Program (2021YFA1502300), the Fundamental Research Funds for the Central Universities (20720190035), and Nanqiang Young Top-notch Talent Fellowship in Xiamen University.

## Author contributions

J.-J.C. and Q.-F.D. conceived and designed the experiments. J.L. analyzed the results and wrote the original draft. X.-X.F. carried out TEM and XPS measurements. J.L., X.-X.F., and T.L. conducted battery-performance measurements. P.X. and Q.H. conducted BET and UV–vis experiments. K.L. provided the structural information of POM clusters. R.-M.Y. contributed to theoretical calculation. M.-S.Z., Q.-F.D., and J.-J.C. participated in discussion and revising the paper.

## Competing interests

The authors declare no competing interests.
