## [Peer Review File · Nature Communications]

REVIEWER COMMENTS

Reviewer #1 (Remarks to the Author):

This paper demonstrates a cobalt centered polyoxometalate (Co-POM) working as a redox catalyst and a host for polysulfide in Li-S battery. This paper discussed well about effect of Co-POM as a catalyst and a host and demonstrated improvement of rate and cycle performance of Li-S battery even though the concept of redox mediator using polyoxometalate (Electrochemistry 84, 882 (2016), J. Am. Chem. Soc. 140, 3134 (2018) ... etc.) and entrapment of polysulfide by transition metal host has been introduced (Nat. Commun. 6, 5682 (2015)).

The data was well organized to draw conclusions. The catalytic performance was quantitatively evaluated and organized with activation energy determined from the reciprocal of charge transfer resistance. The host effect was organized by potentiostatic analysis with support of DFT results. It seems to be helpful for reading to add rate value in Fig. 5a even though space is limited.

The sulfur loading is high 5.6 mg cm⁻² but not dramatically higher than the published literatures (Energy Environ. Sci. 10, 750–755 (2017), Adv. Energy Mater. 10, 2000082 (2020)). One of the significant result might be the cycle performance of 670 mAhg⁻¹@1000cycle with a capacity decay of 0.015%. As far as I have been able to find out, this result is similar to the result in the literature about nano-porous carbon cathode (Nat. Commun. 8, 479 (2017)). Adding a comparison with literature results can be helpful to assess the cycle performance correctly.

Question:

What is the excellence of POM compared with just Co atom without POM embedded in carbon cathode.

Reviewer #2 (Remarks to the Author):

Reviewer's comments:

In the manuscript titled "Single-dispersed {Co₄W₁₈} Molecular Cluster Catalysts for Expedient Sulfur Redox Reactions in Li-S Batteries", the authors reported the use of single-dispersed molecular cluster catalyst (polyoxometalate framework [Co₄(PW₉O₃₄)₂]¹⁰⁻ referred as {Co₄W₁₈}) on reduced graphene oxide (rGO) as electrode material for Li-S batteries. The {Co₄W₁₈}/rGO composite shows enhanced polysulfides (PS) adsorption and also reduced activation energy for PS conversion, thus serving as bifunctional catalyst between the PS conversion and Li₂S precipitation. The material exhibits stable Li-S long-term cycling at higher current density. Overall, the work is interesting and is relevant for the battery community. However, the manuscript lacks in few parts, which need to be further elaborated and discussed. Therefore, I cannot recommend for publication in Nature Communications in the current format. A major revision is recommended for reconsideration.

Below are my specific questions and comments.

1. How much is the poly(ethyleneimine) mass loading on the rGO and how much is the nitrogen content on the modified rGO?
2. How much is the mass loading of the {Co₄W₁₈} on the rGO?
3. In Figure 2c, the authors claim that are significant differences in the EIS spectra and that this is translated into the excellent catalytic properties of the {Co₄W₁₈}/rGO towards polysulfides. However the reviewer is not convinced that the approximately difference between 3 and 6 ohms of the two different materials is reliable to claim the excellent catalytic properties. This difference is in the range of the measuring error and also can varied between different identical cells. Did the authors measure different repetitions of this cells? How much is the deviation between at least 3 identical cells at the same condition? Moreover, this value is dependent on the active electrode surface area. How much is the surface area of the electrodes?
4. The reviewer is curious how good are the fits with this simple equivalent circuit for the EIS spectra in Figure S14 and S15. It is suggested to show the fits for the EIS spectra in these figures.

5. In the figure 4c it is shown the activation energy profiles with the error bars derived from the standard deviation of three independent electrode. The reviewer is curious how this error is also influenced with the error in the fit?
6. There is a misinterpretation of the XPS spectra for the S2p, the two peaks that belong to the sulfite and sulfate in the region between 166 eV and 172 eV are not from the polysulfide interaction but from the lithium triflate salt decomposition on the surface. The similar phenomena is observed with LITFSI (Doron Aurbach et al, J. Electrochem. Soc. 2009, 156, A694). The reviewer suggest to better check the literature on this topic and to change correspondingly this paragraph.
7. How big was the probed area with the XPS? Did 3 different positions on the electrode give different chemical environment? Was the electrode washed or unwashed? Was the electrode transported and inserted inside the XPS in an inert atmosphere or it was on air contact?
8. For the potentiostatic nucleation experiment, the reviewer suggest that the authors also show the corresponding galvanostatic and potentiostatic curves.
9. The reviewer is curious how were the electrodes pretreated for the SEM? Did the authors washed the electrodes to remove the electrolyte salts and polysulfides to visualize the Li₂S deposits? If not, that the SEM images in figure 3a-c and Figure S10 shows only the surface with the present of the polysulfides. It is suggested to wash the electrodes to visualize the crystalline Li₂S and to do the literature overview on this topic. How was the electrode transferred into SEM, was in inert atmosphere?
10. How big are the Li₂S depositions? Are any differences in the size in the case on {Co₄W₁₈}/rGO, {Co₄W₁₈} + rGO and rGO cathodes?
11. Did the authors check the lithium metal electrode how is look-like at high rates (1C, 2C, 5C) and after long-term cycling?
12. How much electrolyte (E/S ratio) was used for the galvanostatic measurements?
13. Coulombic efficiency graphs should use reasonable y-axis scales. For instance, Coulombic efficiency should not be reported on a y-axis scale of 0-100% but rather 90-100%.

Minor issues:

- Is any other better and widely abundant atoms, which can be use instead of Co or Ag for the polyoxometalates for the Li-S battery application?
- Regarding the characteristic absorption peaks on page 6, are this peaks from IR (ATR or transmission mode) or Raman?
- Which DFT model did the authors use for the calculation of the pore size distribution?
- Can the authors show in the supplementary figure 20 also for comparison the capacity vs cycle number for rGO and {Co₄W₁₈} + rGO? Moreover, the reviewer suggest magnifying the CV for the current density between -0.5 and 0.5.
- In the experimental part is missing the chemical name of the water-soluble binder. Is hard to find what is LA 5%.
- Which current collectors were used?
- What was the lithium foil thickness?
- What is the conventional electrolyte?

Reviewer #3 (Remarks to the Author):

This paper describes the synthesis and characterization of a novel catalyst used to enhance Li-S battery performances. The comparison between the bare reduced graphene oxide (rGO), the physical mixture of catalyst and rGO {Co₄W₁₈}+rGO and the composite {Co₄W₁₈}/rGO allowed the authors to unveil the enhanced polysulfides adsorption on the Co sites and the reduced activation energy for polysulfides conversion, factors which contribute to the improved performances of a prototype Li-S pouch cell. The authors analyzed the morphological, chemical and structural changes comparing rGO, {Co₄W₁₈}+rGO and {Co₄W₁₈}/rGO, as well as the electrochemical response of the different

materials during cell operation.

I consider the manuscript well written and the conclusions well supported by the data presented in the main text and in the supporting information. I would suggest the manuscript to be published in Nat. Comm. after some minor revisions. Please consider my comments below.

- 1) Can you please clarify the reference potential? The Ag/AgCl reference electrode is mentioned in the Methods part, but in the text it is not clear if the potential is referred to Li⁺/Li.
- 2) How reproducible are the electrochemical data? Have multiple cells been tested in the same conditions?
- 3) Can you provide more details about the instrumentation used for the XPS (which source has been used, for example) and the XRD measurements (which instrument? which geometry? source energy?)? Concerning the data analysis of the XPS spectra, would it be possible to provide more details about the fitting procedure (background, Gaussian/Lorentzian line shape, FWHM...)?
- 4) While {Co₄W₁₈}/rGO/S cell is shown to provide a high capacity over 100 cycles, the energy density delivered by the cell might be affected by the high loading of {Co₄W₁₈}/rGO in the electrode (in a ratio {Co₄W₁₈}/rGO : S = 3 : 7). Could you please comment on that?
- 5) Figure 1.g: is it possible to make the sketch of the cluster with the same scale as the HAADF-STEM image for better comparison?
- 6) Figure 2.f: it seems that the O 1s peaks are also shifting towards lower BE. Can you please provide an explanation/comment for that?
- 7) Page 12: the link to the Figure 13 in SI is not clear, can you please explain it better?
- 8) Figure 5.g: why is the high S loading cell so unstable? Can you provide an explanation?
- 9) Supplementary Figure 2: can you comment on the presence of K? It does not seem to appear in the materials composition.
- 10) Supplementary Figure 5: can you comment on the dehydrated sample? Why is the background so high? Also, there are differences on the XRD pattern in the rehydrate sample compared to the pristine one. Can you comment them?
- 11) Supplementary Figure 12: CP is not defined in the text/caption.
- 12) Supplementary Figure 13: maybe an inset for each graph would help in seeing the differences in the small features at low frequencies.

Response to the Reviewers' comments:

Reviewer #1: This paper demonstrates a cobalt centered polyoxometalate (Co-POM) working as a redox catalyst and a host for polysulfide in Li-S battery. This paper discussed well about effect of Co-POM as a catalyst and a host and demonstrated improvement of rate and cycle performance of Li-S battery even though the concept of redox mediator using polyoxometalate (Electrochemistry 84, 882 (2016), J. Am. Chem. Soc. 140, 3134 (2018) ... etc.) and entrapment of polysulfide by transition metal host has been introduced (Nat. Commun. 6, 5682 (2015)).

The data was well organized to draw conclusions. The catalytic performance was quantitatively evaluated and organized with activation energy determined from the reciprocal of charge transfer resistance. The host effect was organized by potentiostatic analysis with support of DFT results. It seems to be helpful for reading to add rate value in Fig. 5a even though space is limited.

The sulfur loading is high 5.6 mg cm^{-2} but not dramatically higher than the published literatures (Energy Environ. Sci. 10, 750–755 (2017), Adv. Energy Mater. 10, 2000082 (2020)). One of the significant results might be the cycle performance of 670 mAh g^{-1} @1000cycle with a capacity decay of 0.015%. As far as I have been able to find out, this result is similar to the result in the literature about nano-porous carbon cathode (Nat. Commun. 8, 479 (2017)). Adding a comparison with literature results can be helpful to assess the cycle performance correctly.

Response: *We appreciate the reviewer for the constructive suggestions. Polyoxometalates (POMs) are a series of molecular metal oxide clusters with atomically well-defined composition and chemical structure, which could serve as the ideal molecular models to investigate the structure-reactivity correlations. Therefore, our manuscript focuses on the structure-oriented construction of POM electrocatalysts*

for sulfur redox reactions, and also provides a promising strategy to design advance catalysts for other catalytic fields beyond sulfur conversion reactions.

In order to improve the accessibility of this manuscript, we have rearranged the Fig. 5 and added the detailed rate value in Fig. 5b. Moreover, the specific current values also have been provided to identify the C-rate value in both text and figures, such as “1C corresponding to 1.68 A g⁻¹”. The relevant revision has been made in the section of ‘**Improved electrochemical performance of Li-S batteries**’ which are highlighted in yellow.

We also have added a table in the supplementary information to compare the cycle performance of Li-S batteries with different catalytic host materials. As resulted from Table S1, the {Co₄W₁₈}/rGO/S cathode shows a significant cycle performance of 670 mAh g⁻¹@1000 cycles at 2 C with a capacity decay of 0.015%. And it also exhibits a very competitive performance at high sulfur loading compared to the literature results.

Supplementary Table 1 Comparison of the electrochemical performance of Li-S batteries with different catalytic host materials.

	Sulfur content	Long-term cycling performance			High-sulfur-loading cycling performance		Ref.
		Sulfur loading (mg cm ⁻²)	Capacity after cycling (mAh g ⁻¹)	Capacity decay rate per cycle	Sulfur loading (mg cm ⁻²)	Areal capacity after cycling (mAh cm ⁻²)	
CoP/S	—	1.5	520 (0.5C, 1000th)	0.059%	3	~2.55 (0.5C, 100th)	1
Co-N/G/S	90%	2	681 (1C, 500th)	0.053%	6	5.1 (0.2C, 100 th)	2
Co-Fe-P/S	70%	1	678 (1C, 500th)	0.043%	5.5	3.5 (0.2C, 100th)	3
N-PC@uCo/S	76%	—	780 (1C, 500th)	0.028%	5.9	4.8 (0.2C, 100th)	4
Z-CoS ₂ /S	59%	—	440 (1C, 1000th)	0.04%	2.9	2.2 (0.2C, 150th)	5
CoSA-N-C/S	74.2%	1.2	675 (1C, 1000th)	0.035%	4.9	4.24 (0.2C, 120th)	6
CNTs/Co ₃ S ₄ -NBs/S	70%	—	565 (2C, 500th)	0.068%	3.5	2.87 (0.2C, 150th)	7
Co-NCNT/NP/S	80%	1.2	495 (2C, 800th)	0.047%	3.2	2.7 (0.1C, 200th)	8
CoO/HCN/S	71.3%	1.4	482 (2C, 1000th)	0.043%	3.6	~2.3 (0.5C, 250th)	9

Co ₉ S ₈ /C/S	77%	1.5	560 (2C, 1000th)	0.041%	3	2 (0.5C, 300th)	10
Co/CoP@NC/S	75.8%	2.5	567 (2C, 1000th)	0.03%	4.9	~1.57 (1C, 100th)	11
CMK-3/S@ sulfurized PAN	68%	2.1	598 (2C, 1000th)	0.03%	—	—	12
Fe/Co-N@C/S	70%	~1.5	565 (2C, 1000th)	0.025%	6.4	8.3(0.4mA cm ⁻² , 1th)	13
rGO- CNT/PW ₁₂ @S	76.1%	1.5	555 (3C, 1000th)	0.038%	4.1	3.33 (0.2C, 100th)	14
{Co ₄ W ₁₈ }/rGO/ S	70%	1.5	670 (2C, 1000th)	0.015%	5.6	4.55 (0.1C, 50th)	This work

References

- 1 Qi, C. et al. Cobalt Phosphide Nanoflake-Induced Flower-like Sulfur for High Redox Kinetics and Fast Ion Transfer in Lithium-Sulfur Batteries. *ACS Appl. Mater. Interfaces* 12, 49626-49635 (2020).
- 2 Du, Z. et al. Cobalt in Nitrogen-Doped Graphene as Single-Atom Catalyst for High-Sulfur Content Lithium-Sulfur Batteries. *J. Am. Chem. Soc.* 141, 3977-3985 (2019).
- 3 Chen, Y. et al. Co-Fe Mixed Metal Phosphide Nanocubes with Highly Interconnected-Pore Architecture as an Efficient Polysulfide Mediator for Lithium-Sulfur Batteries. *ACS Nano* 13, 4731-4741 (2019).
- 4 Wang, R. et al. Highly Dispersed Cobalt Clusters in Nitrogen-Doped Porous Carbon Enable Multiple Effects for High-Performance Li-S Battery. *Adv. Energy Mater.* 10, 1903550 (2020).
- 5 Zhang, N. et al. Sulfur encapsulation by MOF-derived CoS₂ embedded in carbon hosts for high-performance Li-S batteries. *J. Mater. Chem. A* 7, 21128-21139 (2019).
- 6 Li, Y. et al. Fast conversion and controlled deposition of lithium (poly)sulfides in lithium-sulfur batteries using high-loading cobalt single atoms. *Energy Storage Mater.* 30, 250-259 (2020).
- 7 Chen, T. et al. Self-Templated Formation of Interlaced Carbon Nanotubes Threaded Hollow Co₃S₄ Nanoboxes for High-Rate and Heat-Resistant Lithium-Sulfur Batteries. *J. Am. Chem. Soc.* 139, 12710-12715 (2017).
- 8 Chen, T. et al. Highly Efficient Retention of Polysulfides in "Sea Urchin"-Like Carbon Nanotube/Nanopolyhedra Superstructures as Cathode Material for

Ultralong-Life Lithium-Sulfur Batteries. Nano Lett. 17, 437-444 (2017).

- 9 *Wu, S. et al. Porous hollow carbon nanospheres embedded with well-dispersed cobalt monoxide nanocrystals as effective polysulfide reservoirs for high-rate and long-cycle lithium-sulfur batteries. J. Mater. Chem. A 5, 17352-17359 (2017).*
- 10 *Chen, T. et al. Metallic and polar Co₉S₈ inlaid carbon hollow nanopolyhedra as efficient polysulfide mediator for lithium-sulfur batteries. Nano Energy 38, 239-248 (2017).*
- 11 *Li, Y. et al. Enhancing Li-S redox kinetics by fabrication of a three dimensional Co/CoP@nitrogen-doped carbon electrocatalyst. Chem. Eng. J. 380, 122595 (2020).*
- 12 *Hu, C. et al. In situ wrapping of the cathode material in lithium-sulfur batteries. Nat. Commun. 8, 479 (2017).*
- 13 *Ye, H. et al. Stepwise Electrocatalysis as a Strategy against Polysulfide Shuttling in Li-S Batteries. ACS Nano 13, 14208-14216 (2019).*
- 14 *Ni, L. et al. Self-Assembled Polyoxometalate Nanodots as Bidirectional Cluster Catalysts for Polysulfide/Sulfide Redox Conversion in Lithium-Sulfur Batteries. ACS Nano 15, 12222-12236 (2021).*

Comment 1: What is the excellence of POM compared with just Co atom without POM embedded in carbon cathode.

Response: As we know, various single Co atom catalysts have been explored for Li-S batteries or other catalytic fields. However, the stabilization of single-atom catalysts in carbon substrate usually requires the specific binding sites, such as heteroatom dopants (e.g., N, P, S, or halogens). And the intricate interactions between single atom and the substrates often make it difficult to assess. Suitable models are thus urgently needed to rationalize the binding-reactivity correlations.

Polyoxometalates (POMs) are a series of atomically well-defined metal oxide clusters which can be considered as the ideal molecular systems to explore the anchoring of active atoms. Moreover, POMs could be directionally designed owing to its structural diversity and compositional adjustability. Thus, POMs show the unique superiority to control the catalytic reactivity by tuning the structure of POM as well as the interactions between POM and active metal atoms.

Reviewer #2: In the manuscript titled “Single-dispersed {Co₄W₁₈} Molecular Cluster Catalysts for Expedition Sulfur Redox Reactions in Li-S Batteries”, the authors reported the use of single-dispersed molecular cluster catalyst (polyoxometalate framework [Co₄(PW₉O₃₄)₂]¹⁰⁻ referred as {Co₄W₁₈}) on reduced graphene oxide (rGO) as electrode material for Li-S batteries. The {Co₄W₁₈}/rGO composite shows enhanced polysulfides (PS) adsorption and also reduced activation energy for PS conversion, thus

serving as bifunctional catalyst between the PS conversion and Li₂S precipitation. The material exhibits stable Li-S long-term cycling at higher current density. Overall, the work is interesting and is relevant for the battery community. However, the manuscript lacks in few parts, which need to be further elaborated and discussed. Therefore, I cannot recommend for publication in Nature Communications in the current format. A major revision is recommended for reconsideration.

Response: We are grateful to the reviewer's positive assessment of our manuscript. And we are also able to address the following specific comments.

Comment 1: How much is the poly(ethyleneimine) mass loading on the rGO and how much is the nitrogen content on the modified rGO?

Response: According to the chemistry formula of the poly(ethyleneimine), (CH₂CH₃NH)_n, the mass loading of PEI could be calculated with the nitrogen element content by the following equation,

$$\omega_{PEI} = \frac{\omega_{element\ N} * M_{PEI}}{M_{element\ N}}$$

Where the ω_{PEI} and $\omega_{element\ N}$ are the mass fraction of the PEI and the element nitrogen, respectively. M_{PEI} and $M_{element\ N}$ are the molar mass of the single-chain PEI (44 g/mol) and the element nitrogen (14 g/mol), respectively.

The nitrogen content on the {Co₄W₁₈}/rGO sample is 2.41 wt% resulted from the element analysis in the Supplementary Fig. 2. And the mass loading of PEI could be determined to be 7.57 wt% (2.41 * 44 / 14). The corresponding content has been indicated in the section of 'Methods—Materials synthesis' which are highlighted in yellow.

Comment 2: How much is the mass loading of the {Co₄W₁₈} on the rGO?

Response: The mass loading of the {Co₄W₁₈} on the rGO could be calculated with the W element content (12.34 wt%) resulted from element analysis in the Supplementary Fig. 2. According to the following equation,

$$\omega_{\{Co_4W_{18}\}} = \frac{\omega_{element\ W} * M_{\{Co_4W_{18}\}}}{18 * M_{element\ W}}$$

Where the $\omega_{\{Co_4W_{18}\}}$ and $\omega_{element\ W}$ are the mass fraction of the {Co₄W₁₈} and the element tungsten, respectively. $M_{\{Co_4W_{18}\}}$ and $M_{element\ W}$ are the molar mass of the {Co₄W₁₈} (4695.86 g/mol) and the element tungsten (183.9 g/mol), respectively.

Therefore, the mass loading of the {Co₄W₁₈} on the rGO could be determined to be 17.5 wt% (12.34 * 4695.86 / 18 / 183.9). The corresponding content has been indicated in the section of 'Methods—Materials synthesis' which are highlighted in yellow.

Comment 3: In Figure 2c, the authors claim that there are significant differences in the EIS spectra and that this is translated into the excellent catalytic properties of the $\{Co_4W_{18}\}/rGO$ towards polysulfides. However, the reviewer is not convinced that the approximately difference between 3 and 6 ohms of the two different materials is reliable to claim the excellent catalytic properties. This difference is in the range of the measuring error and also can vary between different identical cells. Did the authors measure different repetitions of these cells? How much is the deviation between at least 3 identical cells at the same condition? Moreover, this value is dependent on the active electrode surface area. How much is the surface area of the electrodes?

Response: According to the description of symmetrical cells assembly in the section of ‘**Methods—Symmetrical cells assembly and measurement**’, we used the carbon paper disks (diameter of 13 mm) as the electrode collectors with a geometric area of 1.33 cm^2 , but a significantly larger active surface area. Moreover, the electrochemical reaction of higher-order polysulfides in the liquid-phase is relatively favorable, thus leading to the low charge transfer resistances for all the electrodes. However, the deviations of different electrodes could be distinguished which have been affirmed through respectively testing at least three identical cells at the same conditions. As shown in the Supplementary Fig. 10, the difference of the identical cells is small and within the permissible range of experimental error. And the EIS data of different electrodes definitely show the visible reduction of charge transfer resistance after introducing $\{Co_4W_{18}\}$ clusters into the rGO electrodes. The corresponding text and figure have been added in the revised manuscript and supplementary information which are highlighted in yellow.

Supplementary Fig. 10 Nyquist plots of three identical Li_2S_6 symmetrical cells with the (a) rGO, (b) $\{Co_4W_{18}\}+rGO$ and (c) $\{Co_4W_{18}\}/rGO$ electrodes.

Comment 4: The reviewer is curious how good are the fits with this simple equivalent circuit for the EIS spectra in Figure S14 and S15. It is suggested to show the fits for the EIS spectra in these figures.

Response: Thank you for your constructive suggestion. We have rearranged all the EIS spectra, which have been presented in a Nyquist plot with the same scale on X and Y

axes. The equivalent circuit elements shown in Figure S16 were used to fit the impedance spectra. As shown in Figure S17 and S18, the raw data and fitted data were displayed as symbols and lines respectively with good fitting effect.

Supplementary Fig. 17 The raw and fitting impedance data of $\{\text{Co}_4\text{W}_{18}\}/\text{rGO}/\text{S}$ cathodes from 2.7 V to 1.8 V and the corresponding Arrhenius fitting.

Supplementary Fig. 18 The raw and fitting impedance data of rGO/S cathodes from 2.7 V to 1.8 V and the corresponding Arrhenius fitting.

Comment 5: In the figure 4c it is shown the activation energy profiles with the error bars derived from the standard deviation of three independent electrode. The reviewer is curious how this error is also influenced with the error in the fit?

Response: Sorry about our negligence. In detail, the assembled Li-S batteries should be discharged and charged firstly at 1C for 5 cycles before EIS tests. Then the batteries with fully charged state (100% SOC) were placed into the thermostatic chamber with a

specific temperature (263, 278, 293, 308 or 323K) for 2 h. The impedance spectra would be measured at different potentials from 2.7 V to 1.8 V with the same electrode under the particular temperature. Meanwhile, three sets of EIS data were obtained from three independent electrodes, corresponding to three data values of charge transfer resistance at the same conditions. And the average values with standard deviation would be used for linear fitting in Fig. 4b. Thus, the error bars of activation energy in Fig. 4c are synthetically influenced by the average variance of three electrodes and the fitting error. The related introduction has been added in the section of '**Methods—Activation energy of sulfur redox reactions**' which has been highlighted in yellow.

Comment 6: There is a misinterpretation of the XPS spectra for the S2p, the two peaks that belong to the sulfite and sulfate in the region between 166 eV and 172 eV are not from the polysulfide interaction but from the lithium triflate salt decomposition on the surface. The similar phenomena is observed with LITFSI (Doron Aurbach et al, J. Electrochem. Soc. 2009, 156, A694). The reviewer suggest to better check the literature on this topic and to change correspondingly this paragraph.

Response: Thank you for your careful review and your offering of the literature. After reading the reference recommended by the reviewer carefully, we find that the research system of our manuscript is fundamentally different from the literature. Doron Aurbach and his colleagues concluded that the sulfite and sulfate would be formed from the reduction of the lithium triflate salt on the surface of highly active lithium metal.¹ However, in our manuscript, the sulfite and sulfate species would emerge on the surface of {Co₄W₁₈}/rGO materials after interacting with polysulfides solution, in which there is no lithium triflate salt (see the detailed preparation process of polysulfides solution in the section of '**Methods—Polysulfides adsorption test**'). Thus, it is impossible that the sulfite and sulfate species is caused by the decomposition of the lithium triflate.

By referring to the correlational research works, the formation of the sulfite and sulfate species can also be observed on the surface of metal oxides²⁻⁵ or carbon-based materials⁶ after interacting with the polysulfides, which signify the interactions between polysulfides and oxygenic species in the materials. After carefully consideration, we have to insist on the statement that the emergence of sulfite and sulfate species are resulted from the polysulfides interaction with the materials.

References

- 1 Aurbach, D. et al. On the Surface Chemical Aspects of Very High Energy Density, Rechargeable Li-Sulfur Batteries. *J. Electrochem. Soc.* 156, A694-A702 (2009).
- 2 Pang, Q., Kundu, D., Cuisinier, M. & Nazar, L. F. Surface-enhanced redox chemistry of polysulphides on a metallic and polar host for lithium-sulphur batteries. *Nat. Commun.* 5, 4759 (2014).
- 3 Liang, X. et al. A highly efficient polysulfide mediator for lithium-sulfur batteries. *Nat. Commun.* 6, 5682 (2015).

- 4 Zhang, Z. et al. Tantalum-Based Electrocatalyst for Polysulfide Catalysis and Retention for High-Performance Lithium-Sulfur Batteries. *Matter* 3, 920-934 (2020).
- 5 Zhang, H. et al. MXene-Derived Ti_nO_{2n-1} Quantum Dots Distributed on Porous Carbon Nanosheets for Stable and Long-Life Li-S Batteries: Enhanced Polysulfide Mediation via Defect Engineering. *Adv. Mater.* 33, 2008447 (2021).
- 6 Li, G. et al. Stringed “tube on cube” nano hybrids as compact cathode matrix for high-loading and lean-electrolyte lithium-sulfur batteries. *Energy Environ. Sci.* 11, 2372-2381 (2018).

Comment 7: How big was the probed area with the XPS? Did 3 different positions on the electrode give different chemical environment? Was the electrode washed or unwashed? Was the electrode transported and inserted inside the XPS in an inert atmosphere or it was on air contact?

Response: We have added the detailed information about the sample preparation and testing process of XPS in the Supplementary methods. XPS measurements were performed on PHI 5000 Versa Probe II (UIVAC-PHI) using monochromatic Al K α (1486.8 eV) X-ray source at 24 W and 16 KV with a beam spot size of 100 μ m. Definitely, three different positions on the electrode have been chosen to ensure the accuracy and reproducibility of XPS measurements. And all the binding energies have been referenced to the C 1s line at 284.6 eV from adventitious carbon. Moreover, all samples were prepared in the Ar-filled glove box and were washed thoroughly using the anhydrous DME solvent to remove residual polysulfides, and then transferred into the XPS chambers by using the commercial air-isolating containers.

Comment 8: For the potentiostatic nucleation experiment, the reviewer suggest that the authors also show the corresponding galvanostatic and potentiostatic curves.

Response: Thank you for your good suggestion. We have added the corresponding galvanostatic and potentiostatic curves in the Supplementary Fig. 11.

Supplementary Fig. 11 The corresponding galvanostatic and potentiostatic curves of potentiostatic nucleation experiments on the rGO, $\{Co_4W_{18}\}+rGO$, $\{Co_4W_{18}\}/rGO$

electrodes.

Comment 9: The reviewer is curious how were the electrodes pretreated for the SEM? Did the authors washed the electrodes to remove the electrolyte salts and polysulfides to visualize the Li_2S deposits? If not, that the SEM images in figure 3a-c and Figure S10 shows only the surface with the present of the polysulfides. It is suggested to wash the electrodes to visualize the crystalline Li_2S and to do the literature overview on this topic. How was the electrode transferred into SEM, was in inert atmosphere?

Response: *We are convinced that we have washed the electrodes thoroughly using the anhydrous DME solvent to remove the electrolyte salts and polysulfides. The samples were prepared firstly in the glove box filled with Ar gas, and then transferred into the SEM chambers by using the commercial air-isolating containers. As shown in the SEM images from Fig. 3a-c, Supplementary Fig. 12 and 13, the blocky aggregation formed on the electrodes could be determined to be the crystalline Li_2S , rather than the polysulfides, which can be further proved by the corresponding XRD patterns presented in the Supplementary Fig. 14. The similar morphology also has been reported by the previously published works.¹⁻³*

References

- 1 Fan, F. Y., Carter, W. C. & Chiang, Y.-M. Mechanism and Kinetics of Li_2S Precipitation in Lithium-Sulfur Batteries. *Adv. Mater.* 27, 5203-5209 (2015).
- 2 Tao, X. et al. Balancing surface adsorption and diffusion of lithium-polysulfides on nonconductive oxides for lithium-sulfur battery design. *Nat. Commun.* 7, 11203 (2016).
- 3 Wang, R. et al. Bidirectional Catalysts for Liquid-Solid Redox Conversion in Lithium-Sulfur Batteries. *Adv. Mater.* 32, 2000315 (2020).

Comment 10: How big are the Li_2S depositions? Are any differences in the size in the case on $\{\text{Co}_4\text{W}_{18}\}/\text{rGO}$, $\{\text{Co}_4\text{W}_{18}\} + \text{rGO}$ and rGO cathodes?

Response: *As shown in the Fig. 3a-c, Supplementary Fig. 12 and 13, there are significant differences in the size of Li_2S depositions on the rGO , $\{\text{Co}_4\text{W}_{18}\} + \text{rGO}$ and $\{\text{Co}_4\text{W}_{18}\}/\text{rGO}$ electrodes. The blocky Li_2S aggregation would be formed in the rGO electrode, while the size of Li_2S depositions could be reduced obviously by the introduction of the $\{\text{Co}_4\text{W}_{18}\}$ clusters, especially for the $\{\text{Co}_4\text{W}_{18}\}/\text{rGO}$ electrode. This phenomenon reveals that $\{\text{Co}_4\text{W}_{18}\}/\text{rGO}$ could chemically adsorb the polysulfides with the uniformly dispersed active sites, thus achieving its excellent catalytic activity of LiPSs conversion and Li_2S deposition.*

Comment 11: Did the authors check the lithium metal electrode how is look-like at

high rates (1C, 2C, 5C) and after long-term cycling?

Response: Thank you for your useful comment. Due to the missing of the batteries after cyclic test, we have reassembled the $\{Co_4W_{18}\}/rGO/S//Li$ batteries to investigate the surface morphology of the lithium metal electrodes after cycling at different rates (1C, 2C, 5C), which have been washed thoroughly with anhydrous DME solvent. As shown in Supplementary Fig. 22, the deposition of metallic lithium would become rough with the increasing of current density. Moreover, after cycling for 500 cycles at 2C, many black spots appear on the surface of the lithium metal electrodes, indicating the formation of 'dead lithium', which has been believed to be one of the most important reasons for the decline of coulombic efficiency.¹

Supplementary Fig. 22 The optical photographs of lithium metal electrodes after cycling at different rates (1C, 2C, 5C), which have been washed thoroughly with anhydrous DME solvent.

References

- 1 Yan, C., Zhang, X., Huang, J., Liu, Q. & Zhang, Q. Lithium-Anode Protection in Lithium-Sulfur Batteries. *Trends Chem.* 1, 693-704 (2019).

Comment 12: How much electrolyte (E/S ratio) was used for the galvanostatic measurements?

Response: As shown in the section of 'Methods —Li-S batteries assembly and electrochemical characterization', the areal sulfur loading of the cathode was controlled at about 1.5 mg cm^{-2} for the galvanostatic charge-discharge tests. And it would be increased to 3.0, 4.2, 5.6 mg cm^{-2} for high-sulfur-loading cycling test. According to the area of the electrode (diameter of 16 mm) and the amount of electrolyte injected ($50 \mu\text{L}$), E/S ratio could be calculated to be $16.7 \mu\text{L mg}^{-1}$, while 8.3, 5.9, $4.5 \mu\text{L}$

mg^{-1} for the high-sulfur-loading cycling test.

Comment 13: Coulombic efficiency graphs should use reasonable y-axis scales. For instance, Coulombic efficiency should not be reported on a y-axis scale of 0-100% but rather 90-100%.

Response: Thank you for your good comment. We have changed the coulombic efficiency graphs with reasonable y-axis scales as you can see in the Fig. 5.

Comment 14: Is any other better and widely abundant atoms, which can be use instead of Co or Ag for the polyoxometalates for the Li-S battery application?

Response: In recent years, we have been committed to exploiting the polyoxometalates as the active materials or efficient catalysts for possible application in energy storage systems. Owing to its unique structural diversity and compositional adjustability, POMs could be directionally designed. Regarding to the application for the Li-S battery, active Lewis acid-base sites should be constructed in the structural framework of POMs to enhance the chemical interaction with the polysulfides, which has been believed to be an effective strategy for high-performance Li-S battery. For examples, the addition of Co or Ag into the skeleton of POMs could serve as the Lewis acid site due to their unoccupied orbitals that can accept electrons, while the adjacent oxygen atoms can donate electrons that play the roles of Lewis base. Based on the above design principle, various abundant transition metal atoms (Fe, Cu, Zn, et al) could be introduced into the skeleton structure of POMs to construct active Lewis acid-base sites. And further research works are needed to explore more efficient electrocatalysts for Li-S battery application.

Comment 15: Regarding the characteristic absorption peaks on page 6, are this peaks from IR (ATR or transmission mode) or Raman?

Response: Sorry about our carelessness. The characteristic absorption peaks mentioned on page 6 are from IR with the transmission mode, which has been indicated in the revised manuscript.

Comment 16: Which DFT model did the authors use for the calculation of the pore size distribution?

Response: In this manuscript, the pore size distribution was calculated by the Barret-Joyner-Halenda (BJH) method. The detailed information has been provided in the Supplementary methods.

Comment 17: Can the authors show in the supplementary figure 20 also for comparison the capacity vs cycle number for rGO and $\{\text{Co}_4\text{W}_{18}\} + \text{rGO}$? Moreover, the reviewer suggest magnifying the CV for the current density between -0.5 and 0.5.

Response: Thank you for your valuable suggestion. We have assembled the rGO//Li and {Co₄W₁₈}+rGO//Li batteries for galvanostatic charge-discharge and cyclic voltammetry tests. And the results obtained have been added in the Supplementary fig. 23 for comparison with {Co₄W₁₈}/rGO//Li battery. It is concluded that {Co₄W₁₈} clusters would not undergo any electrochemical reactions with high structural stability.

Supplementary Fig. 23 CV curves of (a) rGO//Li, (c) {Co₄W₁₈}+rGO//Li and (e) {Co₄W₁₈}/rGO//Li batteries at the scan rate of 0.1 mV s^{-1} in the voltage range of 1.8~2.7 V. And cycling performance of (b) rGO//Li, (d) {Co₄W₁₈}+rGO//Li and (f) {Co₄W₁₈}/rGO//Li batteries at the current density of 50 mA g^{-1} in the voltage range of 1.8~2.7 V.

Comment 18: In the experimental part is missing the chemical name of the water-soluble binder. Is hard to find what is LA 5%.

Response: Sorry about our negligence. In the preparation of the sulfur-based cathodes, we used LA133 (5 wt%) as the binder which is an aqueous dispersion of acrylonitrile copolymer. The corresponding introduction has been added in the section of 'Methods—Li-S batteries assembly and electrochemical characterization'.

Comment 19: Which current collectors were used? What was the lithium foil thickness? What is the conventional electrolyte?

Response: *In this manuscript, the carbon paper disks (diameter of 16 mm) were used as current collectors for sulfur cathodes. Lithium foils with the thickness of 1 mm were applied as the anodes. And the conventional electrolyte is 0.5 M LiCF₃SO₃ + 0.5 M LiNO₃ dissolved in the mixed solvent of 1,3-dioxolane (DOL) and 1,2-dimethoxyethane (DME) (v/v, 1:1). All these details have been provided in the section of 'Methods—Li-S batteries assembly and electrochemical characterization'.*

Reviewer #3: This paper describes the synthesis and characterization of a novel catalyst used to enhance Li-S battery performances. The comparison between the bare reduced graphene oxide (rGO), the physical mixture of catalyst and rGO {Co₄W₁₈}+rGO and the composite {Co₄W₁₈}/rGO allowed the authors to unveil the enhanced polysulfides adsorption on the Co sites and the reduced activation energy for polysulfides conversion, factors which contribute to the improved performances of a prototype Li-S pouch cell. The authors analyzed the morphological, chemical and structural changes comparing rGO, {Co₄W₁₈}+rGO and {Co₄W₁₈}/rGO, as well as the electrochemical response of the different materials during cell operation.

I consider the manuscript well written and the conclusions well supported by the data presented in the main text and in the supporting information. I would suggest the manuscript to be published in Nat. Comm. after some minor revisions. Please consider my comments below.

Response: *We are glad to the reviewer's positive assessment of our manuscript. And we are also able to revise the manuscript as the reviewer suggest.*

Comment 1: Can you please clarify the reference potential? The Ag/AgCl reference electrode is mentioned in the Methods part, but in the text it is not clear if the potential is referred to Li⁺/Li.

Response: *As mentioned in the section of 'Methods—Catalytic oxidation of Li₂S', the saturated Ag/AgCl reference electrode were applied instead of the lithium foil in the three-electrode linear sweep voltammetry tests due to the methanol solution with 0.1 M Li₂S as the electrolyte. This testing method for catalytic oxidation of Li₂S has been employed by many published research works.¹⁻³ By the way, the potential vs. Ag/AgCl could also be converted to that vs. Li⁺/Li.*

References

- 1 Zhang, Z. et al. Tantalum-Based Electrocatalyst for Polysulfide Catalysis and Retention for High-Performance Lithium-Sulfur Batteries. *Matter* 3, 920-934 (2020).

- 2 Zhang Y. et al. Hierarchical Defective Fe_{3-x}C@C Hollow Microsphere Enables Fast and Long-Lasting Lithium-Sulfur Batteries. *Adv. Funct. Mater.* 30, 2001165 (2020).
- 3 Wang J. et al. Engineering the Conductive Network of Metal Oxide-Based Sulfur Cathode toward Efficient and Longevous Lithium-Sulfur Batteries. *Adv. Energy Mater.* 10, 2002076 (2020).

Comment 2: How reproducible are the electrochemical data? Have multiple cells been tested in the same conditions?

Response: *In this manuscript, at least three cells would be tested in the same conditions to ensure the reproducibility of the electrochemical data.*

Comment 3: Can you provide more details about the instrumentation used for the XPS (which source has been used, for example) and the XRD measurements (which instrument? which geometry? source energy?)? Concerning the data analysis of the XPS spectra, would it be possible to provide more details about the fitting procedure (background, Gaussian/Lorentzian line shape, FWHM...)?

Response: *Sorry about our lack of preciseness. We have provided as much detail as possible about the testing instrument and data analysis, which have been added in the Supplementary methods.*

Comment 4: While {Co₄W₁₈}/rGO/S cell is shown to provide a high capacity over 100 cycles, the energy density delivered by the cell might be affected by the high loading of {Co₄W₁₈}/rGO in the electrode (in a ratio {Co₄W₁₈}/rGO : S = 3 : 7). Could you please comment on that?

Response: *The energy density absolutely would be affected by the high loading of {Co₄W₁₈} catalysts in the electrode. But in our manuscript, the loading of {Co₄W₁₈} catalysts in the electrode is relatively low, despite of the high mass ratio (3 : 7) between {Co₄W₁₈}/rGO and S. According to the slurry component (sulfur composites : Super P : LA133 binder = 7 : 2 : 1) and the mass loading of {Co₄W₁₈} in the {Co₄W₁₈}/rGO materials (17.5 wt%), the actual loading of {Co₄W₁₈} catalysts in the electrode is only 3.7 wt% (=70% * 30% * 17.5%). By the way, the mass proportion of the conductive Super P could be reduced appropriately due to the high conductivity of {Co₄W₁₈}/rGO materials, thus to increase the energy density of the Li-S cell.*

Comment 5: Figure 1.g: is it possible to make the sketch of the cluster with the same scale as the HAADF-STEM image for better comparison?

Response: *Thank you for your good advice. We have added the space-filling structure diagram of the cluster with the same scale as the HAADF-STEM images. As shown in Fig. 1g, the size of the nanoparticles matches well with that of individual {Co₄W₁₈}*

cluster, thus confirming the achievement of monodisperse $\{Co_4W_{18}\}$ clusters throughout the rGO matrix.

Fig. 1g HAADF-STEM image of $\{Co_4W_{18}\}/rGO$ sample

Comment 6: Figure 2.f: it seems that the O 1s peaks are also shifting towards lower BE. Can you please provide an explanation/comment for that?

Response: By checking Fig. 2f carefully, all the O 1s peaks of $\{Co_4W_{18}\}/rGO$ actually shift towards higher binding energy after adsorption with LiPSs corresponding the reduced electron cloud density of oxygen, indicating the interaction between polysulfides and all oxygenic species. As mentioned in our manuscript, the lattice oxygen (O1 peak) and the surface binding oxygen (O2 peak) can act as Lewis bases which could show the “lithium bond”-like interaction with polysulfides. And the adsorbed oxygen (O3 peak) would play an important role for the formation of sulfite and sulfate species which has been mentioned by the reviewer #2, Comment 6. Therefore, the “lithium bond”-like interaction with polysulfides and the adsorbed sulfite and sulfate species on the surface of $\{Co_4W_{18}\}/rGO$ have resulted in the higher binding energy of the O 1s peaks. The related explanation has been provided in the revised manuscript.

Comment 7: Page 12: the link to the Figure 13 in SI is not clear, can you please explain it better?

Response: Thank you for your useful comment. We presume that the reviewer wants to know the detailed process of EIS tests and data analysis for the activation energy of sulfur redox reactions. Hence, we have provided the adequate descriptions in the section of ‘**Methods — Activation energy of sulfur redox reactions**’ and the supplementary methods, which have been highlighted in yellow.

Comment 8: Figure 5.g: why is the high S loading cell so unstable? Can you provide an explanation?

Response: As we know, it is a technical difficulty to prepare the sulfur-based cathodes with high S loading. When the sulfur loading is increased to 5.6 mg cm^{-2} , some cracks would appear on the surface of electrode. Moreover, A large volumetric change (up to 80% expansion) would occur during discharging and charging processes of sulfur

cathodes. Therefore, some active sulfur nanoparticles might detach from the electrode and have no/poor access to electrons and ions, thus leading to the high S loading cell very unstable.

Comment 9: Supplementary Figure 2: can you comment on the presence of K? It does not seem to appear in the materials composition.

Response: As introduced in the section of material preparation, $\{Co_4W_{18}\}/rGO$ materials were synthesized by adding the prepared $K_{10}[Co_4(H_2O)_2(PW_9O_{34})_2] \cdot 22H_2O$ crystals into the aqueous dispersion of the PEI-stabilized rGO. And by the electrostatic interaction, trace K^+ ions would be assembled unavoidably into the $\{Co_4W_{18}\}/rGO$ samples, but couldn't make difference to the catalytic activity of $\{Co_4W_{18}\}/rGO$ materials.

Comment 10: Supplementary Figure 5: can you comment on the dehydrated sample? Why is the background so high? Also, there are differences on the XRD pattern in the rehydrate sample compared to the pristine one. Can you comment them?

Response: The polyoxometalate crystals usually contain some crystal water, and the content of crystal water shows a significant effect on the crystallinity of POMs. Therefore, there are great differences on the XRD pattern in the dehydrated sample compared to the pristine one. But the structure of dehydrated $\{Co_4W_{18}\}$ sample has not been broken, which can be demonstrated by the comparison of FT-IR spectra between the pristine and dehydrated sample as shown in the Supplementary Fig. 5. For the same reason, the content of crystal water in the rehydrate sample has changed, corresponding to a slightly different XRD pattern compared to the pristine sample.

Comment 11: Supplementary Figure 12: CP is not defined in the text/caption.

Response: Sorry about our negligence. 'CP' is the abbreviation of the carbon paper collector.

Comment 12: Supplementary Figure 13: maybe an inset for each graph would help in seeing the differences in the small features at low frequencies.

Response: Thank you for your good suggestion. As the reviewer mentioned, we have added an inset for each EIS graph in the Supplementary Fig. 17 and 18 to improve the accessibility of these data.

REVIEWER COMMENTS

Reviewer #1 (Remarks to the Author):

I am sorry for the additional effort to organize references by Authors. However, references do not show remarkable difference. Also, points of excellence of POM compared to single atom catalyst is still unclear in this manuscript. I cannot recommend for publication in Nature Communications.

Reviewer #3 (Remarks to the Author):

The authors of the manuscripts "Single-dispersed {Co₄W₁₈} Molecular Cluster Catalysts for Expedient Sulfur Redox Reactions in Li-S Batteries" answered my questions and concerns and improved the manuscript.

I would suggest the manuscript to be published in Nature Communications.

Response to the reviewers' comments:

Reviewer #1:

I am sorry for the additional effort to organize references by Authors. However, references do not show remarkable difference. Also, points of excellence of POM compared to single atom catalyst is still unclear in this manuscript. I cannot recommend for publication in Nature Communications.

***Response:** We thank the reviewer for the insightful comments. After referring to the related references, we have clearly illustrated the novelty of polyoxometalates (POMs) material and its performance comparison for Li-S batteries with the state-of-the-art single-atom catalysts (SACs). Besides, we have also done a major revision on the introduction part in the revised manuscript, which we wish to address the concerns raised by the Reviewer on the references organizations.*

More details can be found as following:

1) The novelty of the POMs

As we know, various SACs consisting of atomically-dispersed active metal sites have been exploited for Li-S batteries because of their unique catalytic properties and maximized atom efficiency. However, some crucial challenges remain to be addressed.¹⁻⁵ Due to the high surface free energy, SACs are highly unstable and tend to aggregate, especially with high loading of active metal. Effective supports with specific binding sites should be developed to stabilize the single metal atoms. In this regard, exploring excellent molecular catalysts is important to understand the electrocatalytic behaviors of the intricate interactions and binding modes to bridge the structure-activity gap between experimental and theoretical simulation.

POMs are a variety of atomically well-defined metal oxide clusters which can serve as the ideal molecular systems to explore the anchoring of single metal atoms.⁶ Particularly, the unsaturated binding sites in the lacunary POMs are ideally suitable for permanent active metal atoms coordination and limited leaching and loss of the active metal centers.⁷ The coordination environment and electronic structure of active metal atoms could be determined clearly, which is crucial to gain atomic insights into the fundamental understanding of catalytic mechanism and the corresponding structure-activity relationships. Moreover, POMs offer multifunctionality owing to its fascinating properties, including the multi-electron redox capability and Lewis acidity/basicity, as well as the structural diversity and compositional adjustability.^{8,9} These have been exploited for promising applications in the field of electrocatalysis with unexpected catalytic activity.¹⁰

2) The performance comparison of the POMs with the state-of-the-art SACs

In addition to construct the model catalyst systems, achieving the excellent catalytic performance is another challenge due to the low electrical conductivity and agglomeration of molecular-cluster catalysts. In our work, we developed a single-dispersed {Co₄W₁₈} molecular cluster catalyst embedded on the reduced graphene

oxide (rGO). The remarkable long-term cycling stability of {Co₄W₁₈}/rGO/S cathode with a capacity decay of 0.015% is attributed to the excellent catalytic activity of {Co₄W₁₈}/rGO materials. More importantly, due to the excellent electrocatalytic behaviors, a high areal capacity of 4.55 mAh cm⁻² can be achieved **at a high sulfur loading of 5.6 mg cm⁻² and especially at a low E/S ratio of 4.5 μL mg⁻¹**. Note that a low E/S ratio is very important to construct the high energy density of Li-S batteries.

By the way, after referring to the literatures previously mentioned by the reviewer, strategies to design high-loading sulfur cathodes have been summarized and reviewed (*Adv. Energy Mater.* 10, 2000082 (2020))¹¹, mainly including the functional binders, free-standing electrodes, modified separators, solid electrolytes, as well as the excellent catalytic host materials. For example, a robust biopolymer network binder has been exploited to realize a Li-S battery at a high sulfur loading, nevertheless, the E/S ratio is also high up to 7.8 μL mg⁻¹. (*Energy Environ. Sci.* 10, 750-755 (2017))¹². Thus, the electrocatalytic strategy adopted in our work should be considered to combine with the above approaches for the future design of advanced Li-S batteries.

As suggested by the Reviewer #1, all the comparisons of long-term cycling stability, the battery performance at high sulfur loading and especially low E/S ratio with the state-of-the-art SACs are now summarized in Supplementary Table 1 and Table 2. All the above clarifications and discussions have been indicated in the revised manuscript and supplementary information, which are also highlighted in yellow.

References

- 1 Fang, L., Feng, Z., Cheng, L., Winans, R. E. & Li, T. Design Principles of Single Atoms on Carbons for Lithium-Sulfur Batteries. *Small Methods* 4, 2000315 (2020).
- 2 Wang, F. et al. Single-Atom Electrocatalysts for Lithium Sulfur Batteries: Progress, Opportunities, and Challenges. *ACS Materials Lett.* 2, 1450-1463 (2020).
- 3 Zhuang, Z. et al. Single-atom catalysis enables long-life, high-energy lithium-sulfur batteries. *Nano Res.* 13, 1856-1866 (2020).
- 4 Wang, J. et al. Single Atom-Based Nanoarchitected Electrodes for High-Performance Lithium-Sulfur Batteries. *Adv. Mater. Interfaces* 8, 2002159 (2021).
- 5 Lu, C., Fang, R. & Chen, X. Single-Atom Catalytic Materials for Advanced Battery Systems. *Adv. Mater.* 32, 1906548 (2020).
- 6 Liu, R. & Streb, C. Polyoxometalate-Single Atom Catalysts (POM-SACs) in Energy Research and Catalysis. *Adv. Energy Mater.* 11, 2101120 (2021).
- 7 Ye, J. C. et al. Strategies to Explore and Develop Reversible Redox Reactions of Li-S in Electrode Architectures Using Silver-Polyoxometalate Clusters. *J. Am. Chem. Soc.* 140, 3134-3138 (2018).
- 8 Miras, H. N., Yan, J., Long, D.-L. & Cronin, L. Engineering polyoxometalates with emergent properties. *Chem. Soc. Rev.* 41, 7403-7430 (2012).
- 9 Yang, L. et al. The Intrinsic Charge Carrier Behaviors and Applications of Polyoxometalate Clusters Based Materials. *Adv. Mater.* (2021), DOI: 10.1002/adma.202005019.
- 10 Wang, S.-S. & Yang, G.-Y. Recent Advances in Polyoxometalate-Catalyzed

- Reactions. Chem. Rev. 115, 4893-4962 (2015).*
- 11 Hu, Y. et al. *Strategies toward High-Loading Lithium-Sulfur Battery. Adv. Energy Mater. 10, 2000082 (2020).*
- 12 Liu, J. et al. *Exploiting a robust biopolymer network binder for an ultrahigh-area-capacity Li-S battery. Energy Environ. Sci. 10, 750-755 (2017).*

Reviewer #3:

The authors of the manuscripts "Single-dispersed $\{Co_4W_{18}\}$ Molecular Cluster Catalysts for Expeditious Sulfur Redox Reactions in Li-S Batteries" answered my questions and concerns and improved the manuscript.

I would suggest the manuscript to be published in Nature Communications.

Response: *We appreciate the reviewer for approving the publication of our work. We also thank the reviewer's constructive comments for improving the quality of our work.*

REVIEWERS' COMMENTS

Reviewer #1 (Remarks to the Author):

I appreciate that the authors sincerely answered my question to the manuscripts "Single-dispersed {Co₄W₁₈} Molecular Cluster Catalysts for Expedient Sulfur Redox Reactions in Li-S Batteries"

I agreed the conclusion "When applied for sulfur host materials, {Co₄W₁₈}/rGO/S cathodes could show excellent rate performance and cycling stability which still deliver a specific capacity of about 670 mAh g⁻¹ with a capacity fading of 0.015 % per cycle after 1000 cycles at 2 C."

However, it remains some difficulty on comparison with the established literature.

The newly added Supplementary table 1 and 2 allows comparison between previous reports. Comparative examples using Co catalysts show cycle and rate performance similar to this work. In addition, strict comparison is quite difficult because experimental condition is also different. For example, in Supplementary Table 1, Ref.7 shows that the higher S loading more than 50% allows good retained capacity and decay rate similar to this work the capacity difference is only 7%. Also, in Table 2, Ref 11 and Ref 12 shows that Co catalyst in N doped carbons allows higher S loading and superior cycle performance than this work.

It is thought that even though the performance of Li-S battery using {Co₄W₁₈} shows good performance, it is similar to that of other Co catalysts.

Response to the reviewers' comments:

Reviewer #1:

I appreciate that the authors sincerely answered my question to the manuscripts "Single-dispersed $\{Co_4W_{18}\}$ Molecular Cluster Catalysts for Expeditious Sulfur Redox Reactions in Li-S Batteries"

I agreed the conclusion "When applied for sulfur host materials, $\{Co_4W_{18}\}/rGO/S$ cathodes could show excellent rate performance and cycling stability which still deliver a specific capacity of about 670 mAh g⁻¹ with a capacity fading of 0.015 % per cycle after 1000 cycles at 2 C."

However, it remains some difficulty on comparison with the established literature.

The newly added Supplementary table 1 and 2 allows comparison between previous reports. Comparative examples using Co catalysts show cycle and rate performance similar to this work. In addition, strict comparison is quite difficult because experimental condition is also different. For example, in Supplementary Table 1, Ref.7 shows that the higher S loading more than 50% allows good retained capacity and decay rate similar to this work the capacity difference is only 7%.

Also, in Table 2, Ref 11 and Ref 12 shows that Co catalyst in N doped carbons allows higher S loading and superior cycle performance than this work.

It is thought that even though the performance of Li-S battery using $\{Co_4W_{18}\}$ shows good performance, it is similar to that of other Co catalysts.

***Response:** We appreciate the reviewer's approval for the performance of $\{Co_4W_{18}\}/rGO/S$ cathodes. And we have tried our best to make a fair comparison with the previous reports. All the electrochemical performance comparisons in the Supplementary Table 1 and 2 are based on the Li-S coin cells at room temperature using the conventional ether-based electrolyte (DOL/DME, 1:1, by volume). However, the strict comparison is really difficult due to the different experimental condition. For example, Ref. 7 in Supplementary Table 1 used the Li_2S_8 cathode instead of S cathode, which could improve the sufficient interaction between the active polysulfide species and the catalytic materials, thus leading to the good retention capacity and low decay rate.*

Also, we have to mention again that our manuscript focuses on constructing the model molecular-catalyst systems for Li-S batteries to fundamentally understand the catalytic mechanism and the structure-activity relationship. Subsequently, achieving the good performance of Li-S battery using $\{Co_4W_{18}\}$ is already a breakthrough despite the absence of prominent superiority compared with the previously reported Co-based catalysts.